# Smooth Flipping Probability for Differentially Private Sign Random Projection Methods

**Ping Li,   Xiaoyun Li**

LinkedIn Ads
700 Bellevue Way NE, Bellevue, WA 98004, USA
{pingli98, lixiaoyun996}@gmail.com

## Abstract

We develop a series of differential privacy (DP) algorithms from a family of random projection (RP) and sign random projection (SignRP) methods. We first show how to improve the previous DP-RP approach using the "optimal Gaussian mechanism". Then, we propose a series of DP-SignRP algorithms that leverage the robustness of the "sign flipping probability" of random projections. That is, given $x = \sum_{i=1}^{p} u_i w_i$ where $u$ is a $p$-dimensional data vector and $w$ is a symmetric random vector, $sign(x)$ only has a fairly small probability to be flipped if there is a small modification on data $u$, depending on the specific distribution of $w$. This robustness leads to our novel design of "smooth flipping probability" for SignRP-type algorithms with better utility than using the standard randomized response mechanism. Retrieval and classification experiments demonstrate that, among the presented DP-RP algorithms, **DP-SignOPORP** (where OPORP is an improvement over the celebrated count-sketch algorithms), performs the best in general.

In the industrial practice, DP methods were not very popular for machine learning or search, largely because the performance typically would drop substantially if DP is applied. Since our proposed new DP algorithms have significantly improved the performance, it is anticipated that our work will motivate a wide adoption of DP in practice. Finally, we stress that, since our methods are applied to the original data (i.e., feature vectors), the privacy of downstream tasks is naturally protected.

## 1   Introduction

Protecting data privacy has become an urgent need and a trending research topic recently. Among many notions of privacy, the "differential privacy" (DP) [30] has gained tremendous attention in both the research community and industrial applications, and has been widely applied to numerous tasks such as frequency estimation [32, 26], clustering [36, 34], regression and classification [14, 85], DP-SGD [1, 35], principle component analysis [40], empirical risk minimization [15], graph analysis [33], matrix completion [8, 45], etc. In a machine learning model, there are several choices on when to deploy differential privacy (DP): (a) at the data collection/processing stage [84, 81, 5, 19]; (b) during the model training stage [1, 78]; and (c) on the model output (or summary statistics) [29]. Among them, applying DP as early as at the data collection stage provides strong protection in the sense that the subsequent operations or outputs will become private by the post-processing property of DP. Moreover, when the data holder needs to release the data to third-parties, a privacy-preserving data publishing mechanism (at an early stage) becomes necessary.

In this paper, we focus on the differential privacy of the broad family of random projection (RP) methods, including RP, sign random projections (SignRP) and count-sketch-type algorithm. Our algorithms can be applied to data publishing and processing for downstream machine learning tasks.

37th Conference on Neural Information Processing Systems (NeurIPS 2023).

## 1.1 Random Projections (RP) and Sign Random Projections (SignRP)

In practice, data compression and dimension reduction techniques can be crucial when dealing with massive (high-dimensional) data. The random projection (RP) method is an important and fundamental dimensionality reduction algorithm. Denote $u \in \mathbb{R}^p$ as the data vector with $p$ features. With some $k$ (and typically $k \ll p$), we define the random projection of $u$ as

$$X = \frac{1}{\sqrt{k}} W^T u, \ \ W \in \mathbb{R}^{p \times k}. \tag{1}$$

The entries of the random matrix $W$ typically follow the Gaussian distribution or Gaussian-like distribution such as the Rademacher (symmetric Bernoulli) distribution. It has been well-understood that, with a sufficient number of projections ($k$), the distance between two vectors (using the same projection $W$) is preserved within a small multiplicative error with high probability. The dimensionality reduction and geometry preserving properties make RP widely useful in numerous applications, such as distance estimation, nearest neighbor search, clustering, classification, compressed sensing, permutation recovery, etc. [46, 44, 22, 6, 2, 37, 24, 76, 10, 28, 56, 38, 23, 21, 60, 68, 75, 82, 83].

While RP is able to reduce the dimensionality, storing and transmitting projected data might still be expensive for large datasets. One can further compress the RPs by quantization/discretization, where we only use a few bits to represent the projected values [41, 9, 27, 89, 54, 58, 50, 55, 61, 62]. In this paper, we will consider the differential privacy of the extreme case of sign (1-bit) random projection (SignRP), also known as the "SimHash" [13, 24], where only the sign of the projected data is stored. The signs still preserve the angle/cosine information. SignRP has been widely used for approximate near neighbor (ANN) search [39] by building hash tables from the bits [44, 13, 71]. RP and SignRP can also be extended to non-linear random features [69, 63, 64].

## 1.2 Count-Sketch and OPORP: Efficient RP-type Alternatives

The celebrated count-sketch [12] can be viewed as a highly efficient RP-type algorithm, because, as an option, it only requires "one permutation + one random projection" (OPORP) [57], as opposed to $k$ projections in RP. Applications of count-sketch include graph embedding [80], word & image embedding [16, 87, 3, 72, 86], model & communication compression [79, 17, 70, 42, 59], etc. The recent work by [57] improved the original count-sketch in two aspects: (i) using a fixed-length binning scheme makes the algorithms more convenient and also reduces the variance by a factor of $\frac{p-k}{p-1}$; (ii) the projected data (i.e., vectors in $k$ dimensions) should be normalized before they are used to estimate the original cosine $\rho$, which also substantially reduces the estimation variance, essentially from $(1 + \rho^2)/k$ (un-normalized) to $(1 - \rho^2)^2/k$ (normalized).

**The main contributions** of this work include the following:

- In Section 3, for DP-RP, we first revisit the prior work [49] on the $(\epsilon, \delta)$-DP random projection based on Gaussian noise addition (called DP-RP-G). Then we incorporate the optimal Gaussian mechanism and propose an improved method DP-RP-G-OPT. We also develop DP-OPORP based on the optimal Gaussian mechanism.

- In Section 4, we propose two algorithms for privatizing the (1-bit) DP-SignRP. The first method, DP-SignRP-RR, is based on the standard "randomized response" (RR). Then, we propose an improved method named DP-SignRP-RR-smooth based on a proposed concept of *"smooth flipping probability"*, thanks to the robustness brought by the "aggregate-and-sign" operations of SignRP. Finally, we extend the idea of smooth flipping probability to OPORP and show that DP-SignOPORP is more advantageous in terms of privacy.

- In Section 5, we conduct retrieval and classification experiments on benchmark datasets. For full-precision methods, the proposed DP-OPORP substantially improves prior method DP-RP-G. For the 1-bit RP variants, the suggested DP-SignOPORP can outperform DP-RP and DP-OPORP especially when $\epsilon$ is not large (e.g., $\epsilon \leq 5$). Our results also verify the advantage of smooth flipping probability over the classic bit flipping strategy.

## 2   Background on Differential Privacy

Throughout the paper, $\| \cdot \|_1$ and $\| \cdot \|_2$ are the $l_1$ and $l_2$ norms, respectively, and $\| \cdot \|$ will denote the $l_2$ norm if there no risk of confusion. Let $u \in \mathcal{U}$ be the data vector of a user where $\mathcal{U}$ denotes the data domain. In this paper, we consider real-valued data vectors with bounded range, i.e., $\mathcal{U} = [-1, 1]^p$ for convenience. We also assume $\|u\| > 0$, i.e., there is no all-zero data sample. The formal definition of differential privacy (DP) is given as follows.

**Definition 2.1** (Differential Privacy [30]). *For a randomized algorithm $\mathcal{M} : \mathcal{U} \mapsto Range(\mathcal{M})$, if for any two adjacent datasets $u$ and $u'$, it holds that*

$$Pr[\mathcal{M}(u) \in O] \leq e^\epsilon Pr[\mathcal{M}(u') \in O] + \delta \tag{2}$$

*for $\forall O \subset Range(\mathcal{M})$ and some $\epsilon, \delta \geq 0$, then algorithm $\mathcal{M}$ is called $(\epsilon, \delta)$-differentially private. If $\delta = 0$, $\mathcal{M}$ is called $\epsilon$-differentially private.*

In this work, we follow the standard setup in the literature of DP sketching/hashing (e.g., [7, 49, 73, 74, 88, 25, 65]) that neighboring data vectors $u$ and $u'$ only differ in one dimension, which is usually referred to as the "attribute-level DP"—an adversary cannot detect a small change in any attribute of $u$ based on the output of the algorithm.

**Definition 2.2** ($\beta$-adjacency). *Let $u \in [-1, 1]^p$ be a data vector. A vector $u' \in [-1, 1]^p$ is said to be $\beta$-adjacent to $u$ if $u'$ and $u$ differ in one dimension $i$, and $|u_i - u_i'| \leq \beta$.*

Data vectors $u$ and $u'$ satisfying Definition 2.2 are called $\beta$-adjacent or $\beta$-neighboring. In the literature (e.g., [49]), $\beta = 1$ is typically used. We allow a general $\beta$ to accommodate the practical needs. Moreover, the concept of sensitivity is important for the design of DP noise addition mechanisms.

**Definition 2.3** ($l_2$-sensitivity). *Let $Nb(u)$ denote the neighbor set of $u$, and $\mathcal{N}(\mathcal{U}) = \{(u, u') : u' \in Nb(u), u, u' \in \mathcal{U}\}$ be the collection of all possible neighboring data pairs. The $l_2$-sensitivity of a function $f : \mathcal{U} \mapsto \mathbb{R}^k$ is defined as $\triangle_2 = \max_{(u,u') \in \mathcal{N}(\mathcal{U})} \|f(u) - f(u')\|_2$.*

The following composition theorem of DP is also useful in our analysis.

**Theorem 2.1** (Composition Theorem [31]). *Let $\mathcal{M}_j : \mathcal{U} \to \mathbb{R}$ be an $(\epsilon_j, \delta_j)$-DP algorithm for $j = 1, ..., k$. Then $\mathcal{M}(u) = (\mathcal{M}_1(u), ..., \mathcal{M}_k(u))$ is $(\sum_{j=1}^k \epsilon_i, \sum_{j=1}^k \delta_j)$-DP.*

## 3   Revisiting Differentially Private Random Projection Methods

We first revisit the DP noise addition mechanism for random projections and show how to apply the optimal Gaussian mechanism [4] to improve prior methods. We also propose a DP algorithm based on a variant of count-sketch which is computationally much more efficient than dense projections.

### 3.1   Gaussian Noise Mechanism for DP-RP

Let $\mathcal{N} = \{u, u' \in \mathcal{U} : u \in Nb(u)\}$ be the collection of all possible $\beta$-neighboring data pairs, where $Nb(u)$ is the neighbor set of $u$. For a $p$-by-$k$ projection matrix $W$, the $l_2$-sensitivity of the RP (1), according to Definition 2.3, can be precisely computed by

$$\triangle_2 = \frac{1}{\sqrt{k}}\beta \max_{(u,u') \in \mathcal{N}(\mathcal{U})} \|W^T u - W^T u'\|_2 = \frac{1}{\sqrt{k}}\beta \max_{i=1,...,p} \|W_{[i,:]}\|_2, \tag{3}$$

where $W_{[i,:]}$ denotes the $i$-th row of $W$. As such, the DP noise level depends on the choice of projection matrix $W$. A comparison of Rademacher vs. Gaussian projection in DP-RP method can be found in Section B.1. With this calculation, the general Gaussian noise mechanism for DP-RP is summarized in Algorithm 1. We present the algorithms for a single data point $u$, and the procedure is applied to every data vector in the database (with same $W$ but independent noise). After random projection, we simply add a random Gaussian noise vector following iid $N(0, \sigma^2)$ to the projected data. The following "DP-RP-G" result is known in the literature.

**Theorem 3.1** (DP-RP-G [49]). *Let $\triangle_2$ be defined in (3). For any $\epsilon > 0$ and $0 < \delta < \frac{1}{2}$, DP-RP-G in Algorithm 1 is $(\epsilon, \delta)$-DP if $\sigma \geq \triangle_2 \frac{\sqrt{2(\log(1/\delta)+\epsilon)}}{\epsilon}$.*

---
**Algorithm 1:** DP-RP-G and DP-RP-G-OPT
---
1 **Input:** Data $u \in [-1,1]^p$, privacy parameters $\epsilon > 0$, $\delta \in (0,1)$, number of projections $k$
2 **Output:** $(\epsilon, \delta)$-differentially private random projections $\tilde{x} \in \mathbb{R}^k$
3 Apply RP $x = \frac{1}{\sqrt{k}} W^T u$, where $W \in \mathbb{R}^{p \times k}$ has iid entries from $N(0,1)$ or Rademacher
4 Compute the sensitivity $\triangle_2$ by (3)
5 Generate the random noise vector $G \in \mathbb{R}^k$ whose entries are iid samples from $N(0, \sigma^2)$ where $\sigma$
 is obtained by Theorem 3.1 (DP-RP-G) or Theorem 3.2 (DP-RP-G-OPT)
6 Return $\tilde{x} = x + G$
---

**Remark 3.1.** *Note that in implementation, $\triangle_2$ is computed by (3) using the realization of the projection matrix $W$. In Appendix A, we provide a high probability bound on $\triangle_2$ and more analysis and discussion on the Laplace mechanism for DP-RP.*

In Theorem 3.1 (as well as the classical Gaussian mechanism, e.g., [31]), the analysis on the noise level is based on upper bounding the tail of Gaussian distribution. It can be improved by computing the exact tail probabilities. [4, Theorem 8] proposed an optimal Gaussian mechanism based on this idea, which allows us to improve Theorem 3.1 and obtain the DP-RP-G-OPT method as follows.

**Theorem 3.2** (DP-RP-G-OPT). *Suppose $\triangle_2$ is defined as (3). For any $\epsilon > 0$ and $0 < \delta < 1$, DP-RP-G-OPT in Algorithm 1 achieves $(\epsilon, \delta)$-DP if $\sigma \geq \sigma^*$ where $\sigma^*$ is the solution to the equation*

$$\Phi\left(\frac{\triangle_2}{2\sigma} - \frac{\epsilon\sigma}{\triangle_2}\right) - e^\epsilon \Phi\left(-\frac{\triangle_2}{2\sigma} - \frac{\epsilon\sigma}{\triangle_2}\right) = \delta, \tag{4}$$

*where $\Phi(\cdot)$ is the cdf of the standard normal distribution.*

**Remark 3.2.** *For both Theorem 3.2 and Theorem 3.1, the Gaussian noise level is $\sigma = \mathcal{O}(\frac{\triangle_2}{\epsilon})$ when $\epsilon \to 0$, and $\sigma = \mathcal{O}(\frac{\triangle_2}{\sqrt{\epsilon}})$ when $\epsilon \to \infty$, which is rate optimal [4]. Essentially, Theorem 3.2 reduces the noise variance of Theorem 3.1 by analyzing the tail probability exactly.*

### 3.2 DP-OPORP: A More Efficient Alternative

The standard random projections (1) require $k$ projections. We can reduce the cost from $O(kp)$ to $O(p)$ by count-sketch: "binning + Rademacher RP". Basically, we split the data entries into $k$ bins, and apply RP in each bin to generate $k$ samples. [88] studied noise injection mechanism for count-sketch [12]. Recently, [57] proposed OPORP (One Permutation + One Random Projection), an improved variant of count-sketch using fixed-length binning and the normalized estimator. The steps are summarized in Algorithm 2. Note that the output $x_i$ can be $l_2$ normalized to reduce variance.

---
**Algorithm 2:** OPORP: count-sketch with fixed-length binning
---
1 **Input:** Data vector $u \in \mathbb{R}^p$; Number of projected samples $k$
2 **Output:** $k$ OPORP samples
3 Apply a permutation $\Pi : [p] \mapsto [p]$ to $u$ to get $u_\Pi$
4 Split the $p$ permuted data columns into $k$ consecutive length-$p/k$ bins: $u_\Pi = [u_\Pi^{(1)}, ..., u_\Pi^{(k)}]$
5 Generate a vector $w \in \mathbb{R}^p$ following Rademacher distribution, denoted as $w = [w^{(1)}, ..., w^{(k)}]$
6 Return $k$ projected samples by $x_i = w^{(i)^T} u_\Pi^{(i)}$, for $i = 1, ..., k$.
---

For OPORP, the sensitivity $\triangle_2 = \beta$, due to the binning and Rademacher projection: changing one coordinate of $u$ by $\beta$ leads to a change of $\beta$ in term of $l_2$ distance between the projected vectors.

**Theorem 3.3** (DP-OPORP). *Adding noise from iid $N(0, \sigma^2)$ to OPORP (the output of Algorithm 2) achieves $(\epsilon, \delta)$-DP, where $\sigma$ is the solution to (4) with $\triangle_2 = \beta$.*

The advantages of DP-RP and DP-OPORP over the approach of adding Gaussian noise to raw data for inner product estimation are theoretically justified in Appendix C.

# 4 DP-SignRP: Differentially Private Sign Random Projections

We now propose our main algorithms that output SignRP with DP guarantees. Firstly, we analyze the standard randomized response (RR) technique, DP-SignRP-RR, under Gaussian random projections. Next, we propose the concept of "smooth flipping probability" and develop DP-SignRP-RR-smooth as an improvement. Finally, we propose DP-SignOPORP that combines all the techniques.

## 4.1 DP-SignRP-RR by Randomized Response

---

**Algorithm 3:** DP-SignRP-RR

---

1 **Input:** Data $u \in [-1, 1]^p$; $\epsilon > 0$, $0 < \delta < 1$; Number of projections $k$; norm lower bound $m$

2 **Output:** Differentially private sign random projections

3 Apply RP by $x = \frac{1}{\sqrt{k}} W^T u$, where $W \in \mathbb{R}^{p \times k}$ is a random $N(0, 1)$ matrix

4 Let $N_+(m, \delta, k, p)$ be computed as in Proposition 4.3

5 Compute $\tilde{s}_j = \begin{cases} sign(x_j), & \text{with prob. } \frac{e^{\epsilon'}}{e^{\epsilon'}+1} \\ -sign(x_j), & \text{with prob. } \frac{1}{e^{\epsilon'}+1} \end{cases}$ for $j = 1, ..., k$, with $\epsilon' = \epsilon/N_+(m, \delta, k, p)$

6 Return $\tilde{s}$ as the DP-SignRP of $u$

---

We develop DP-SignRP-RR based on the classic randomized response (RR) mechanism [77, 31]. As summarized in Algorithm 3, after we apply random projection $x = W^T u$, we take $s = sign(x)$. Then, for each $s_j$, we keep the sign with probability $\frac{e^{\epsilon'}}{e^{\epsilon'}+1}$ and flip the sign with probability $\frac{1}{e^{\epsilon'}+1}$. Here $\epsilon' = \epsilon/N_+$, where $N_+(m, \delta, k, p)$ is an upper bound on the number of different signs (among $k$ signs) of $x = W^T u$ and $x = W^T u'$ for any $\beta$-adjacent $(u, u')$, which will be derived later in Proposition 4.3. Here, $m$ is a lower bound on the $l_2$ norm of the data, i.e., $\|u\| \geq m$ for all $u \in \mathcal{U}$. The final output is $\tilde{s}$ after perturbing $s$ by the above procedure.

**Theorem 4.1.** *Algorithm 3 is $(\epsilon, \delta)$-DP.*

*Proof.* For any data point $u \in \mathcal{U}$ and its $\beta$-adjacent neighbor $u'$, denote $s = sign(W^T u) \in \{-1, +1\}^k$, $s' = sign(W^T u') \in \{-1, +1\}^k$, and let $\tilde{s}$ and $\tilde{s}'$ be the corresponding randomized sign vectors output by Algorithm 3. Denote $S = \{i : s_j \neq s'_j\}$ and $S^c = [k] \setminus S$. For any vector $y \in \{-1, +1\}^k$, define $S_0 = \{j \in S : s_j = y_j\}$, $S_1 = \{j \in S : s_j \neq y_j\}$, $S_0^c = \{j \in S^c : s_j = y_j\}$ and $S_1^c = \{j \in S^c : s_j \neq y_j\}$. By Proposition 4.3, we know that the event $\{|S| \leq N_+(m, \delta, k, p)\}$ happens with probability at least $1 - \delta$. In this event, we have

$$\log \frac{Pr(\tilde{s} = y)}{Pr(\tilde{s}' = y)} = \log \frac{\prod_{j \in S_0^c} \frac{e^{\epsilon'}}{e^{\epsilon'}+1} \prod_{j \in S_1^c} \frac{1}{e^{\epsilon'}+1} \prod_{j \in S_0} \frac{e^{\epsilon'}}{e^{\epsilon'}+1} \prod_{j \in S_1} \frac{1}{e^{\epsilon'}+1}}{\prod_{j \in S_0^c} \frac{e^{\epsilon'}}{e^{\epsilon'}+1} \prod_{j \in S_1^c} \frac{1}{e^{\epsilon'}+1} \prod_{j \in S_0} \frac{1}{e^{\epsilon'}+1} \prod_{j \in S_1} \frac{e^{\epsilon'}}{e^{\epsilon'}+1}}$$

$$\leq \log \frac{\prod_{j \in S} \frac{e^{\epsilon'}}{e^{\epsilon'}+1}}{\prod_{j \in S} \frac{1}{e^{\epsilon'}+1}} = |S|\epsilon' \leq N_+\epsilon' = \epsilon.$$

Since this event occurs with probability at least $1 - \delta$, the overall procedure is $(\epsilon, \delta)$-DP. $\square$

### 4.1.1 The flipping probability and calculation of $N_+$

In the proof of Theorem 4.1, $N_+$ is a upper bound on $S = \{i : s_j \neq s'_j\}$ where $s = sign(W^T u)$ and $s' = sign(W^T u')$ for neighboring data $(u, u')$, which determines the flipping probability thus the utility. Next, we analyze $N_+$. The following is a useful lemma; all proofs are placed in Appendix E.

**Lemma 4.2.** *Let $X_1, ..., X_p$ be iid $N(0, \sigma_x^2)$ variables. Let $Y \sim N(0, \sigma_y^2)$ be another Gaussian random variable with arbitrary dependence structure with $X_i$'s. Let $r = \sigma_x/\sigma_y \leq 1$. Then*

$$P_+(r, p) := Pr(\max_{i=1,...,p} |X| > |Y|) \leq \int_0^\infty 2p[2\Phi(t) - 1]^{p-1}[2\Phi(rt) - 1]\phi(t)dt, \quad (5)$$

*where $\phi(x)$ and $\Phi(x)$ are the standard Gaussian pdf and cdf, respectively.*

Equipped with Lemma 4.2, we can derive $N_+$ in Algorithm 3, an upper bound on the number of projected signs that are possible to change when $u$ is replaced by any neighboring data vector $u'$.

**Proposition 4.3** (Bound $N_+$). *Suppose $u \in [-1,1]^p$ and $\beta \leq \|u\|$. Denote $r = \frac{\beta}{\|u\|}$ and $F_{\|u\|,p} = P_+(\frac{\beta}{\|u\|}, p)$ in (5). Denote $s = sign(W^T u)$ and $s' = sign(W^T u')$ for $\beta$-neighboring $(u, u')$, and $S = \{i : s_j \neq s'_j\}$. Then with probability $1 - \delta$, $|S| \leq N_+(\|u\|, \delta, k, p) = B^{-1}(\delta; F_{\|u\|,p}, k)$, where $B^{-1}(\delta; F_{\|u\|,p}, k)$ represents the inverse cdf of Binomial distribution with rate $F_{\|u\|,p}$ and $k$ trials.*

Since $F_{\|u\|,p} = P_+(r, p)$ is an increasing function in $r = \beta/\|u\|$, $N_+$ would be smaller if $\beta/\|u\|$ is smaller. Therefore, in DP-SignRP-RR, the probability of flipping the true SignRP, $\frac{1}{e^{\epsilon/N_+}+1}$, would be smaller when the data norm (or, its lower bound $m$) is large compared with $\beta$.

### 4.1.2 Utility in angle estimation by DP-SignRP-RR

Define the DP-SignRP-RR estimator of the angle between two data points $u$ and $v$ as

$$\hat{\theta}_{RR} = \pi(1 - \hat{P}_{RR}), \quad \text{where } \hat{P}_{RR} = \frac{(e^{\epsilon'} + 1)^2}{(e^{\epsilon'} - 1)^2} \frac{1}{k} \sum_{j=1}^{k} \mathbb{1}\{\tilde{s}_{1j} = \tilde{s}_{2j}\} - \frac{2e^{\epsilon'}}{(e^{\epsilon'} - 1)^2}. \quad (6)$$

**Theorem 4.4.** *Let $\rho = \cos(u, v)$ and $\theta = \cos^{-1}(\rho)$. Run Algorithm 3 with $N_+$ given in Proposition 4.3 and define the DP-SignRP-RR angle estimator by (6). We have $\mathbb{E}[\hat{\theta}_{RR}] = \theta$. As $k \to \infty$, $\hat{\theta}_{RR} \to N(\theta, \frac{V_{RR}}{k})$, with*

$$V_{RR} = \theta(\pi - \theta) + \frac{2\pi^2 e^{\epsilon/N_+}}{(e^{\epsilon/N_+} - 1)^2} + \frac{4\pi^2 e^{2\epsilon/N_+}}{(e^{\epsilon/N_+} - 1)^4}.$$

Theorem 4.4 says that $\hat{\theta}_{RR}$ is an unbiased estimator of $\theta$ and asymptotically normal. Compared with $\frac{\theta(\pi-\theta)}{k}$, the variance of the cosine estimator from non-DP SignRP [41, 13], we see that $\hat{\theta}_{RR}$ incurs an extra variance (i.e., utility loss) of $\frac{\pi^2}{k}\left[\frac{2e^{\epsilon/N_+}}{(e^{\epsilon/N_+}-1)^2} + \frac{4e^{2\epsilon/N_+}}{(e^{\epsilon/N_+}-1)^4}\right]$. This quantity increases as $\epsilon$ gets smaller, illustrating the utility-privacy trade-off of DP-SignRP-RR.

**Optimal projection dimension $k^*$.** Since by Proposition 4.3 we know that $N_+ \asymp F_{\|u\|,p} k$ roughly, the term $\left[\frac{2e^{\epsilon/N_+}}{(e^{\epsilon/N_+}-1)^2} + \frac{4e^{2\epsilon/N_+}}{(e^{\epsilon/N_+}-1)^4}\right]$ would increase with $k$. Therefore, there exists an optimal $k^*$ that minimizes the estimation variance. When $k$ is sufficiently large, we have

$$\frac{V_{RR}}{k} \approx \frac{\theta(\pi - \theta)}{k} + \frac{2\pi^2 F_{\|u\|,p}^2 k}{\epsilon^2} + \frac{8\pi^2 F_{\|u\|,p}^4 k^3}{\epsilon^4},$$

using the approximation that $e^x - 1 \approx x$ when $x$ is small. To find the optimal $k$ minimizing this expression, we compute the derivative and set it as zero:

$$-\frac{\theta(\pi - \theta)}{k^2} + \frac{2\pi^2 F_{\|u\|,p}^2}{\epsilon^2} + \frac{24\pi^2 F_{\|u\|,p}^4 k^2}{\epsilon^4} = 0 \implies k^* \asymp \frac{\epsilon \theta(\pi - \theta)}{F_{\|u\|,p}}.$$

This analysis suggests the optimal $k^*$ of DP-SignRP-RR is larger when: (1) $\epsilon$ is large; (2) $F_{\|u\|,p}$ is small, which is typically true when the norm of the data is relatively large.

### 4.2 Smooth Flipping Probability: the Benefit of Robustness

Next, we improve DP-SignRP-RR based on a novel tool which we call *"smooth flipping probability"*. The idea is inspired by the "smooth sensitivity" for DP noise addition [66], which essentially says that, the farther $u$ is from a high "local sensitivity" region, the less noise is needed for $u$ to achieve DP. The local sensitivity at a specific data point $u$ is defined as $LS(u) = \max_{u' \in Nb(u)} \|f(u) - f(u')\|_2$, which differs from Definition 2.3 in that local sensitivity only considers the local neighbors of $u$.

What is the "local perturbation" for flipping the SignRP? Consider one projection $x_j = W[:, j]^T u$ and $s_j = sign(x_j)$. For a neighbor $u'$ of $u$, denote $x'_j = W[:, j]^T u'$ and $s'_j = sign(x'_j)$. By $\beta$-adjacency, $s'_j$ is possible to be different from $s_j$ only if $|x_j| < \beta \max_{i=1,...,p} |W_{ij}|$. This implies:

- When $|x_j| < \beta \max_{i=1,...,p} |W_{ij}|$, the "local sign flipping" is required, since $s'_j$ may differ from $s_j$. We can apply the standard RR: keep $s_j$ with prob. $\frac{e^{\epsilon/k}}{e^{\epsilon/k}+1}$ and flip otherwise.

- When $|x_j| \geq \beta \max_{i=1,...,p} |W_{ij}|$, there does not exist $u' \in Nb(u)$ such that $s'_j \neq s_j$. Thus, the "local sign flipping" is not needed. However, if we do not perturb $s_j$, DP would be violated since $s'_j$ might need random flipping. Instead, we use a *"smooth flipping probability"* which applies less perturbation for points far away from the "high perturbation region", i.e., the regime where $|x_j| < \beta \max_{i=1,...,p} |W_{ij}|$. Specifically, we define $L_j = \lceil \frac{|x_j|}{\beta \max_{i=1,...,p} |W_{ij}|} \rceil$. The probability of keeping the sign $s_j$ is $P_j^{(u)} = \frac{e^{\epsilon'_j}}{e^{\epsilon'_j}+1}$ where $\epsilon'_j = \frac{L_j}{k}\epsilon$. It can be shown (e.g., in the proof of Theorem 4.5) that this flipping probability is "smooth": for $\forall u, u'$ that are neighbors, $P_j^{(u)} \leq e^{\epsilon/k} P_j^{(u')}$, which justifies its name.

---

**Algorithm 4:** DP-SignRP-RR-smooth using smooth flipping probability

1 **Input:** Data $u \in [-1,1]^p$; $\epsilon > 0$; Number of projections $k$

2 **Output:** Differentially private sign random projections

3 Apply RP by $x = W^T u$, where $W \in \mathbb{R}^{p \times k}$ is a random $N(0,1)$ matrix

4 Compute $L_j = \lceil \frac{|x_j|}{\beta \max_{i=1,...,p} |W_{ij}|} \rceil$ for $j = 1, ..., k$

5 Compute $\tilde{s}_j = \begin{cases} sign(x_j), & \text{with prob. } \frac{e^{\epsilon'_j}}{e^{\epsilon'_j}+1} \\ -sign(x_j), & \text{with prob. } \frac{1}{e^{\epsilon'_j}+1} \end{cases}$ for $j = 1, ..., k$, with $\epsilon'_j = \frac{L_j}{k}\epsilon$

6 Return $\tilde{s}$ as the DP-SignRP of $u$

---

The concrete steps of DP-SignRP-RR-smooth are summarized in Algorithm 4. Note that, the above two cases can be unified into one, since $\frac{e^{\epsilon/k}}{e^{\epsilon/k}+1}$ is the smooth flipping probability with $L_j = 1$. In Figure 1, we provide an illustrative example of the smooth flipping probability (red) and the "local flipping probability" (green) discussed above. The global upper bound (black dash) corresponds to the standard randomized response. The local flipping probability is only non-zero at $L = 1$. The proposed smooth flipping probability is non-zero for all $L > 0$, but keeps shrinking exponentially as $L$ gets larger, i.e., as the projected value is farther from $0$.

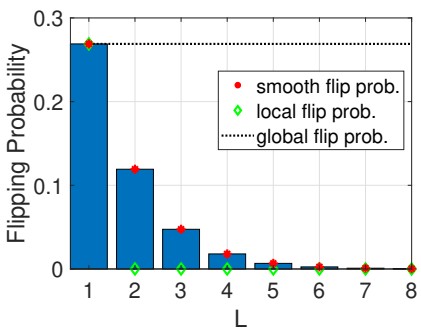

Figure 1: The smooth flipping prob., local flipping prob., and the global upper bound. $\epsilon = 1$, $k = 1$. $L$ is from Algorithm 4.

**Theorem 4.5.** *Algorithm 4 achieves $\epsilon$-DP.*

**Comparison with DP-SignRP-RR (Algorithm 3).** In terms of algorithm design and privacy, DP-SignRP-RR-smooth (Algorithm 4) has two advantages over the standard DP-SignRP-RR method:

1. DP-SignRP-RR-smooth does not require (assume) a lower bound $m$ on the data norms;

2. DP-SignRP-RR-smooth achieves $\epsilon$-DP, while DP-SignRP-RR only guarantee $(\epsilon, \delta)$-DP.

Note that Algorithm 3 can also achieve $\epsilon$-DP if we set the flipping probability to be $\frac{1}{e^{\epsilon/k}+1}$ (for all projected values). This corresponds to the global upper bound (black dotted line) in Figure 1. Thus, DP-SignRP-RR-smooth requires much smaller flipping probability than DP-SignRP-RR.

**DP-SignRP with Rademacher projection.** Similar to DP-RP, the flipping probabilities also depend on the projection matrix $W$ through $N_+$ (for DP-SignRP-RR) and $L$ (for DP-SignRP-RR-smooth). In Appendix B.2, we analytically show that Rademacher RP requires smaller flipping probability than that of Gaussian RP for both variants.

### 4.3 DP-SignOPORP with Smooth Flipping Probability

We incorporate the smooth flipping technique to develop DP-SignOPORP, as given in Algorithm 5 with two variants. DP-SignOPORP-RR is based on the standard randomized response (RR), while DP-SignOPORP-RR-smooth is an improved version with our proposed smooth flipping probability.

---

**Algorithm 5:** DP-SignOPORP-RR and DP-SignOPORP-RR-smooth

---

1 **Input:** Data $u \in [-1, 1]^p$; $\epsilon > 0$; Number of projections $k$
2 **Output:** Differentially private sign OPORP
3 Apply Algorithm 2 with a random Rademacher projection vector to get the OPORP $x$
   **DP-SignOPORP-RR:**
4 Compute $\tilde{s}_j = \begin{cases} sign(x_j), & \text{with prob. } \frac{e^\epsilon}{e^\epsilon + 1} \\ -sign(x_j), & \text{with prob. } \frac{1}{e^\epsilon + 1} \end{cases}$ for $j = 1, ..., k$
   **DP-SignOPORP-RR-smooth:**
5 Compute $L_j = \lceil \frac{|x_j|}{\beta} \rceil$ for $j = 1, ..., k$
6 Compute $\tilde{s}_j = \begin{cases} sign(x_j), & \text{with prob. } \frac{e^{\epsilon'_j}}{e^{\epsilon'_j} + 1} \\ -sign(x_j), & \text{with prob. } \frac{1}{e^{\epsilon'_j} + 1} \end{cases}$ for $j = 1, ..., k$, with $\epsilon'_j = L_j \epsilon$
7 For $\tilde{s}_j = 0$, assign a random coin in $\{-1, 1\}$
8 Return $\tilde{s}$ as the DP-SignRP of $u$

---

**Benefit of binning.** In Algorithm 5, the key difference compared with DP-SignRP methods is that, the $\frac{1}{N_+}$ (in Algorithm 3) or $\frac{1}{k}$ (in Algorithm 4) is removed from the flipping probability formulas, which is a significant advantage in privacy. This improvement is a result of the binning step. By Definition 2.2, two neighboring data $u'$ and $u$ differ in one coordinate. For SignRP, since each projected data is an aggregation of the whole data vector, when we switch from $u$ to $u'$, (in principle) all $k$ projections are possible to change. In contrast, in OPORP, since each data entry appears in only one bin, $u'$ will only cause exactly one projected output to change, which enhances privacy.

**Theorem 4.6.** *Both variants in Algorithm 5 are $\epsilon$-DP.*

*Proof.* The proof follows the idea of the proofs of DP-SignRP methods, but we need to additionally consider the empty bins. Denote $x$ and $x'$ as the OPORP from Algorithm 2 using a same random vector $w$, and let $s = sign(x)$, $s' = sign(x')$. Suppose $u$ and $u'$ differ in dimension $i$, and dimension $i$ is assigned to the $j^*$-th bin with $j^* = \lceil \pi(i)/(p/k) \rceil$, where $\pi : [p] \mapsto [p]$ is the permutation in OPORP. For any $y \in \{-1, 1\}^k$, it is easy to see that to compute $\log \frac{Pr(\tilde{s}=y)}{Pr(\tilde{s}'=y)}$, it suffices to look at the $j^*$-th output sample because other probabilities cancel out. For the $j^*$-th sample, when $s_{j^*} \neq 0$ and $s'_{j^*} \neq 0$, by the same arguments as in the proof of Theorem 4.5, we know that $e^{-\epsilon} \leq \frac{Pr(s_{j^*})=a}{Pr(s'_{j^*})=a} \leq e^\epsilon$, $a \in \{-1, 1\}$, for both variants (DP-SignOPORP-RR and DP-SignOPORP-RR-smooth). When one of $s_{j^*}$ and $s'_{j^*}$ equals 0, it is also easy to see that (assume $s_{j^*} = 1$ and $s'_{j^*}$ is a random coin)

$$1 \leq \frac{Pr(s_{j^*}) = 1}{Pr(s'_{j^*}) = 1} = \frac{2e^\epsilon}{e^\epsilon + 1} \leq e^\epsilon, \quad e^{-\epsilon} \leq \frac{Pr(s_{j^*}) = -1}{Pr(s'_{j^*}) = -1} = \frac{2}{e^\epsilon + 1} \leq 1.$$

This holds for both variants. In particular, this case corresponds to $L_{j^*} = 1$ for the smooth flipping probability. This proves the theorem. $\square$

**Multiple repetitions.** In Algorithm 5 Line 7, if OPORP generates an empty bin (i.e., $x_j = 0$), we must assign a random sign to it in order to maintain DP. This tends to undermine the utility since a random coin does not provide any useful information about the data. To avoid too many empty bins, one simple way is to repeat the DP-SignOPORP (with smaller $k$) for $t > 1$ times, and concatenate the output vectors [12]. For example, if the target $k = 256$, we may run Algorithm 5 for $t = 4$ times, each time with $256/4 = 64$ projected values and privacy budget $\epsilon/4$. We still obtain a length-256 bit vector with $\epsilon$-DP by the composition theorem. In the experiments, we will see that this strategy may improve the overall performance of DP-SignOPORP.

# 5 Experiments

We present experiments on retrieval and classification tasks to demonstrate the effectiveness of our proposed methods. We compare the following algorithms:

- Raw-data-G-OPT: adding optimal Gaussian noise with sensitivity $\beta$ to the original data, which is the most basic DP method for privatizing the original data.
- DP-RP-G (Algorithm 1 + Theorem 3.1): Gaussian DP-RP with the Gaussian mechanism.
- DP-RP-G-OPT (Algorithm 1 + Theorem 3.2): DP-RP with Gaussian random projection matrix and the optimal Gaussian noise.
- DP-OPORP (Theorem 3.3): one permutation + one random projection with Rademacher projection vector and optimal Gaussian noise.
- DP-SignOPROP-RR and DP-SignOPORP-RR-smooth (Algorithm 5): signed OPORP with standard randomized response and with smooth flipping probability, respectively.

As noted in Section 4.3, due to binning, DP-SignOPORP is substantially better than DP-SignRP. Thus we present results for DP-SignOPORP in our experiments (and only $\beta = 1$ for conciseness).

**Similarity Search.** We first test the methods in similarity search problems, using two standard image retrieval datasets, MNIST [53] and CIFAR [51]. For both datasets, we treat the training set as the database, and use the samples from the test set as the queries. We set the ground truth neighbors for each query as the top-50 sample vectors with the highest cosine similarity to the query. To search with DP-RP (and DP-OPORP), we estimate the cosine between a query and a sample point by the cosine between their corresponding output noisy projected values. We report only the precisions (and please see Appendix for other metrics), averaged over 10 repetitions.

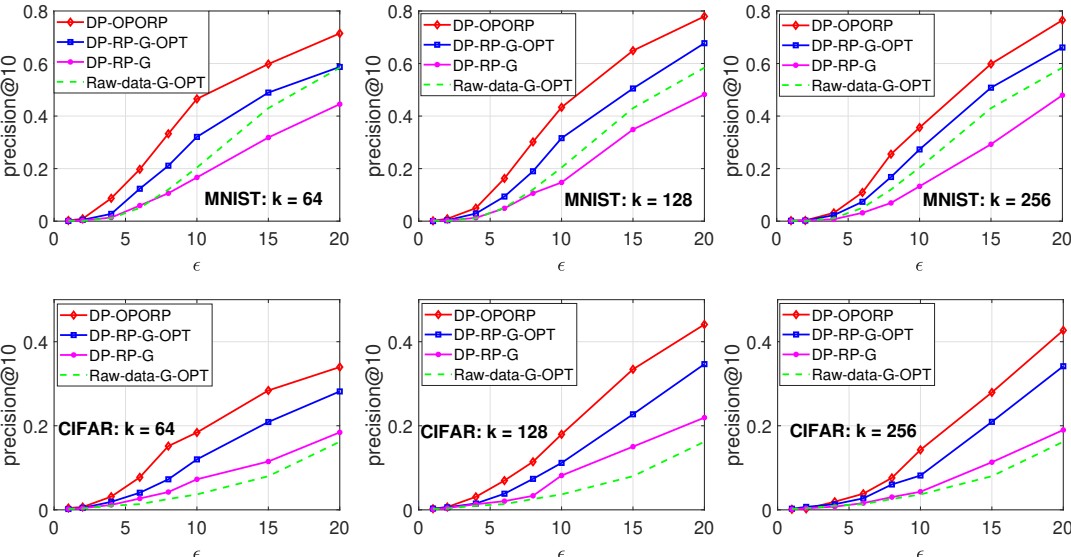

Figure 2: Retrieval recall and precision on MNIST and CIFAR, $\beta = 1$, $\delta = 10^{-6}$.

Figure 2 shows that, as a result of the optimal Gaussian mechanism, DP-RP-G-OPT substantially outperforms the prior method in the literature, DP-RP-G, at all $\epsilon$. DP-OPORP achieve better performance than DP-RP-G-OPT and is recommended as the best (full-precision) DP-RP variant. These two methods provide much higher precision and recall than the optimal Gaussian mechanism applied to the original data (Raw-data-G-OPT).

In Figure 3, we plot the precision and recall of DP-SignOPORP-RR (with standard randomized response), and DP-SignOPORP-RR-s (with smooth flipping probability). We present the results with repetition $t = 2, 4$ which yield good overall performance. As we can see, (i) the proposed smooth flipping probability considerably improves the standard randomized response technique, and (ii)

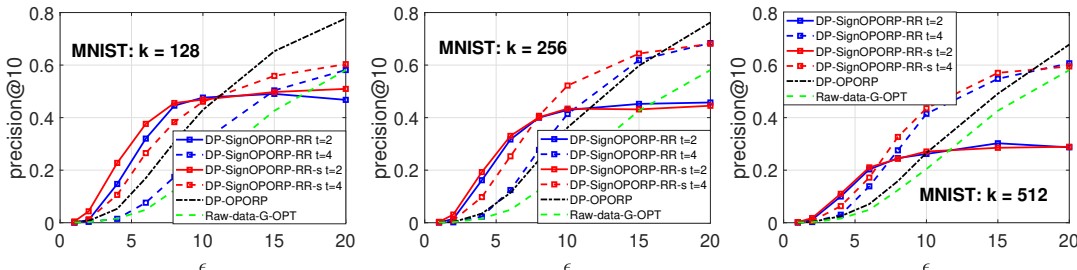

Figure 3: Retrieval on MNIST with DP-SignOPORP-RR and DP-SignOPORP-RR-smooth. For DP-OPORP, DP-SignOPORP-RR, and Raw-data-G-OPT, $\delta = 10^{-6}$.

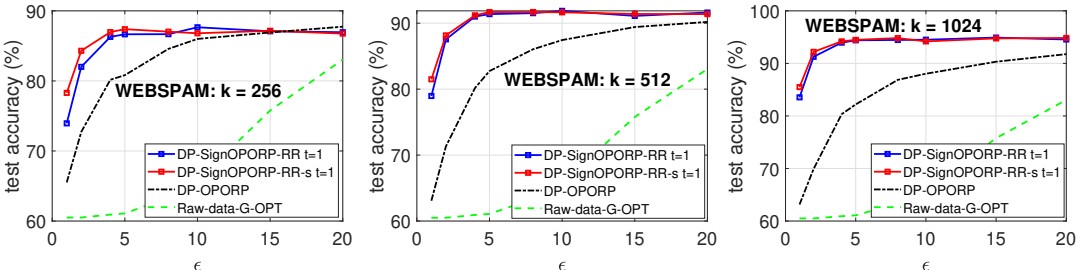

Figure 4: SVM test accuracy on WEBSPAM. For $(\epsilon, \delta)$-DP methods, $\delta = 10^{-6}$.

DP-SignOPORP in general provides better search accuracy than DP-OPORP when $\epsilon < 10 \sim 15$. This range of $\epsilon$ is common in the use cases of DP for providing reasonable privacy protection. Also, we note that smaller $t$ (repetitions) typically performs better with smaller $\epsilon$ but worse when $\epsilon$ is large.

**Classification.** We evaluate the performance of DP-SignOPORP in classification problems trained by SVM [20], on the WEBSPAM dataset [11]. We apply the "max normalization", i.e., divide each data column by its largest magnitude such that the data entries are bounded in $[-1, 1]$. The SVM results are presented in Figure 4. For this task, we plot DP-SignOPORP with $t = 1$. We again observe the advantage of DP-SignOPORP over DP-OPORP, and the advantage of the proposed smooth flipping probability over the standard randomized response. Specifically, when $\epsilon \approx 5$, the test accuracy of Raw-data-G-OPT is around 60% (for WEBSPAM, this is almost the same as random guessing), but DP-SignOPORP can achieve 95% classification accuracy with $k = 1024$.

# 6 Conclusion

For a database $U \in \mathbb{R}^{n \times p}$ consisting of $n$ data points, our goal is to make its random projections (RP) and SignRP differentially private—the DP guarantee we have derived holds independently and simultaneously for every data vector $u \in U$. There are works on randomized algorithms that assume the randomness of the algorithm are "internal" and also kept private [7, 73, 25]. We assume the projection matrix $W$ is known/public, which is stronger in privacy and enables broader applications of the released data. Also, a "known projection matrix" is typical in the local DP setting [47, 18], since the projection matrix is created and shared among the users by the data aggregator.

In our study, we first revisit the prior work [49] on the $(\epsilon, \delta)$-DP random projection based on Gaussian noise addition ("DP-RP-G"). Then we incorporate the optimal Gaussian mechanism and propose an improved method DP-RP-G-OPT. We also develop DP-OPORP based on the optimal Gaussian mechanism. We then propose algorithms for privatizing the (1-bit) DP-SignRP: "DP-SignRP-RR" based on the standard "randomized response" (RR), and "DP-SignRP-RR-smooth" based on a proposed concept of "smooth flipping probability" of SignRP. We extend the idea of smooth flipping probability to OPORP (a variant of count-sketch) and demonstrate the advantage of DP-SignOPORP.

Experiments on retrieval and classification tasks justify the effectiveness of our proposed methods. In particular, DP-SignOPORP outperforms other methods especially when $\epsilon$ is not large (e.g., $\epsilon \leq 5$). These results also verify the advantage of using the smooth flipping probability.

## Acknowledgement

The authors would like to thank several researchers from Microsoft Research: Janardhan Kulkarni, Yin Tat Lee, Sergey Yekhanin, for the helpful discussions.

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

# Contents

# A    Analytic DP-RP and the Laplace Mechanism

## A.1    Analytic DP-RP-G Approach: Bounding the Sensitivity

According to Theorem 3.1, one will choose $\sigma$ based on the observed $\Delta_2$ which is calculated according to (3) for a given realization of the projection matrix $W$. For the convenience of theoretical analysis, it is often helpful to derive the criterion for choosing $\sigma$ directly in terms of the input parameters $\epsilon$, $\delta$, $p$, and $k$, to understand and better compare the noise level. We call the corresponding algorithm to be "Analytic DP-RP-G". This is achieved by bounding $\Delta_2$ in high probability.

**Lemma A.1** ([52]). *Let $Y_1, ..., Y_n$ be iid standard Gaussian random variables, and denote $Z_i = \sum_{i=1}^{n} Y_i^2$. Then for any $t > 0$,*

$$Pr(Z \geq n + 2\sqrt{nt} + 2t) \leq \exp(-t).$$

**Lemma A.2** (Bounding $\Delta_2$). *Let $W \in \mathbb{R}^{p \times k}$ be a random matrix with iid $N(0,1)$ entries, and $\Delta_2$ be defined in (3). Then for any $0 < \delta < 1$,*

$$Pr\left(\Delta_2 \geq \beta\sqrt{1 + 2\sqrt{\log(p/\delta)/k} + 2\log(p/\delta)/k}\right) \leq \delta.$$

*Proof.* Let $e_i$ be the unit base vector with the $i$-th dimension being $1$ and all other dimensions being zero. For each $i = 1, ..., p$, $W^T e_i$ follows iid chi-square distribution with $k$ degree of freedom. Applying Lemma A.1 we have that for any $\delta' > 0$,

$$Pr\left(\|W^T e_i\|^2 \geq \beta^2(k + 2\sqrt{k\log(1/\delta')} + 2\log(1/\delta'))\right) \leq \delta', \ \ \forall i = 1, ..., p.$$

Applying union bound gives

$$Pr\left(\frac{1}{k}\beta^2 \max_{i=1,...,p} \|W^T e_i\|^2 \geq 1 + 2\sqrt{\log(1/\delta')/k} + 2\log(1/\delta')/k\right) \leq p\delta'.$$

Setting $\delta = p\delta'$ and taking the square root on both sides completes the proof. $\qquad\square$

With the high probability bound on the $l_2$-sensitivity $\Delta_2$, we have the following result.

**Theorem A.3** (Analytic DP-RP-G). *For any $\epsilon > 0$ and $0 < \delta < \frac{1}{2}$, Algorithm 1 achieves $(\epsilon, \delta)$-DP when $\sigma \geq \frac{\beta\sqrt{1 + 2\sqrt{\log(2p/\delta)/k} + 2\log(2p/\delta)/k}\sqrt{2(\log(2/\delta)+\epsilon)}}{\epsilon}$.*

*Proof.* By Lemma A.2, we know that with probability $\delta/2$, $\Delta_2$ is bounded by $\beta\sqrt{1 + 2\sqrt{\log(2p/\delta)/k} + 2\log(2p/\delta)/k}$. In this event, by Theorem 3.1, Algorithm 1 is $(\epsilon, \delta/2)$-DP when $\sigma \geq \Delta_2 \frac{\sqrt{2(\log(2/\delta)+\epsilon)}}{\epsilon}$. Therefore, the approach is $(\epsilon, \delta)$-DP. $\qquad\square$

We remark that, since this analytic version is built upon a high probability bound on the sensitivity, in most cases, the noise variance in Theorem A.3 will be slightly larger than the one computed by the realized/exact sensitivity in Theorem 3.1.

## A.2    The Laplace Noise Mechanism for DP-RP

We briefly discuss the Laplace mechanism for DP-RP (and why we did not use it).

**Definition A.1** ($l_1$-sensitivity). *Let $Nb(u)$ denote the neighbor set of $u$, and $\mathcal{N}(\mathcal{U}) = \{(u, u') : u' \in Nb(u), u, u' \in \mathcal{U}\}$ be the collection of all possible neighboring data pairs. The $l_1$-sensitivity of a function $f : \mathcal{U} \mapsto \mathbb{R}^k$ is defined as $\triangle_1 = \max_{(u,u') \in \mathcal{N}(\mathcal{U})} \|f(u) - f(u')\|_1$.*

**Theorem A.4** (Laplace mechanism [30]). *Given any function $f : \mathcal{U} \mapsto \mathbb{R}^k$ and $\epsilon > 0$, the randomized algorithm $\mathcal{M}(u) = f(u) + (Z_1, ..., Z_k)$ is $\epsilon$-DP, where $Z_1, ..., Z_k$ follow iid $Laplace(\Delta_1/\epsilon)$. The centered Laplace distribution with scale $\lambda$ has density function $g(x) = \frac{1}{2\lambda}e^{-\frac{|x|}{\lambda}}$.*

---
**Algorithm 6:** DP-RP-L
---
1 **Input:** Data $u \in [-1,1]^p$, privacy parameters $\epsilon > 0$, number of projections $k$
2 **Output:** $\epsilon$-differentially private random projections $\tilde{x} \in \mathbb{R}^k$
3 Apply RP $x = \frac{1}{\sqrt{k}} W^T u$, where $W \in \mathbb{R}^{p \times k}$ is a random $N(0,1)$ matrix
4 Compute sensitivity $\Delta_1$ by Definition A.1
5 Generate iid random noise vector $L \in \mathbb{R}^k$ following $Laplace(\frac{\Delta_1}{\epsilon})$
6 Return $\tilde{x} = x + L$
---

By Theorem A.4, one may add $Laplace(\lambda)$ noise to the projected data with $\lambda = \Delta_1/\epsilon$ to achieve $\epsilon$-DP. For completeness, we detail the approach in Algorithm 6, which is the same as Algorithm 1 except for the noise added. $\Delta_1$ is the $l_1$-sensitivity for a given $W$ given in Definition A.1: $\Delta_1 = \frac{1}{\sqrt{k}} \beta \max_{i=1,\dots,p} \|W_{[i,:]}\|_1$. Each term in the maximum operator is a sum of $k$ iid standard half-normal random variables ($Y = |X|$ where $X$ is standard normal). The readers might be interested in a comparison between the noise level of the two noise addition mechanisms, i.e., DP-RP-G and DP-RP-L. To observe the dependence of the noise level $\lambda$ on $\epsilon$, $k$ and $p$, we derive a tail bound on the $l_1$-sensitivity. We first establish the following lemma.

**Lemma A.5** (Half-normal tail bound). *Let $X_1, ..., X_n$ be iid $N(0,1)$ random variables and let $Y_i = |X_i|, \forall i$. Denote $Z = \sum_{i=1}^n Y_i$. Then for any $t > 0$, it holds that*

$$Pr(Z \geq \sqrt{2n^2 \log 2 + 2nt}) \leq \exp(-t).$$

*Proof.* By the Markov inequality, we write (with some $\zeta > 0$)

$$Pr(Z \geq t) = Pr(\exp(\zeta \sum_{i=1}^n Y_i) \geq \exp(\zeta t)) \leq \frac{\mathbb{E}[\exp(\zeta \sum_{i=1}^n Y_i]}{\exp(\zeta t)}.$$

By the independence of $Y_i$ and the moment generating function of half-normal distribution, we have

$$\mathbb{E}[\exp(\zeta \sum_{i=1}^n Y_i] = \prod_{i=1}^n \mathbb{E}[\exp(\zeta Y_i)] = \exp(\frac{n\zeta^2}{2})(2\Phi(\zeta))^n = \exp(\frac{n\zeta^2}{2} + n \log 2\Phi(\zeta))$$

$$\leq \exp(\frac{n\zeta^2}{2} + n \log 2).$$

Hence, we have

$$Pr(Z \geq t) \leq \exp(\frac{n\zeta^2}{2} - t\zeta + n \log 2),$$

which is minimized at $\zeta = \frac{t}{n}$, leading to

$$Pr(Z \geq t) \leq \exp(-\frac{t^2}{2n} + n \log 2).$$

Taking $t = \sqrt{2n^2 \log 2 + 2nt}$ completes the proof. $\qquad \square$

Using Lemma A.5, we obtain a high probability bound on the $l_1$-sensitivity as follows. The proof is omitted since it is similar to the proof of Lemma A.2.

**Lemma A.6** (Bounding $\Delta_1$). *Let $W \in \mathbb{R}^{p \times k}$ be a random matrix with iid $N(0,1)$ entries, and $\Delta_1$ be defined in Definition A.1. Then for any $0 < \delta < 1$,*

$$Pr\left(\Delta_1 \geq \beta \sqrt{2k \log 2 + 2 \log(p/\delta)}\right) \leq \delta.$$

It is important to point out that, unlike in the Gaussian mechanism, simply inserting the upper bound in Lemma A.6 to replace the (exact) $\Delta_1$ in the noise level of $Laplace(\Delta_1/\epsilon)$ in Algorithm 6 does not provide pure $\epsilon$-DP. In other words, we must compute the actual $\Delta_1$ by Definition A.1 for a realization

of the projection matrix $W$. This is because, using the high probability bound of $\Delta_1$ would introduce a failure probability (i.e., when the actual $\Delta_1$ is larger than the bound) which violates $\epsilon$-DP.

**Comparison with DP-RP-G-OPT.** Lemma A.6 provides an estimate of the noise level required by the Laplace mechanism. We can roughly compare DP-RP-L with DP-RP-OPT in terms of the noise variances, which equal $\sigma^2$ for Gaussian and $2\lambda^2$ for Laplace, respectively. Lemma A.6 shows that in most cases, the Laplace noise requires $\lambda = \mathcal{O}(\frac{\sqrt{k}}{\epsilon})$. Recall that in DP-RP-G and DP-RP-G-OPT, the Gaussian noise level $\sigma = \mathcal{O}(\frac{1}{\epsilon})$ when $\epsilon \to 0$ and $\sigma = \mathcal{O}(\frac{1}{\sqrt{\epsilon}})$ when $\epsilon \to \infty$. Thus, the Gaussian noise would be smaller with relatively small $\epsilon$, and the Laplace noise would be smaller only when $\epsilon = \Omega(k)$, which is usually too large to be useful since in practice $k$ is usually hundreds to thousands. As a result, we would expect DP-RP-G methods to perform better than DP-RP-L in common scenarios. Therefore, in this work, we mainly focused on the Gaussian mechanism for DP-RP noise addition.

# B    Comparison of Different Projection Matrices and the Benefits of Rademacher RP

Besides Gaussian random projection, we can also adopt other types of projection matrices which might even work better for DP. The following distributions of $w_{ij}$ are popular:

- The uniform distribution, $\sqrt{3} \times unif[-1, 1]$. The $\sqrt{3}$ factor is placed here to have $\mathbb{E}(w_{ij}^2) = 1$ by following the convention in the practice of random projections.
- The "very sparse" distribution, as used in [56]:

$$w_{ij} = \sqrt{s} \times \begin{cases} -1 & \text{with prob.} \quad 1/(2s) \\ 0 & \text{with prob.} \quad 1 - 1/s, \\ +1 & \text{with prob.} \quad 1/(2s) \end{cases} \tag{7}$$

  which generalizes [2] (for $s = 1$ and $s = 3$). Note that when $s = 1$, it is also called the "symmetric Bernoulli" distribution or the "Rademacher" distribution.

Next, we compare these various types of projection matrices and show that Rademacher (symmetric Bernoulli) random projection is superior to Gaussian random projection for both DP-RP and DP-SignRP in that less perturbation is required to achieve the same privacy level.

## B.1    Rademacher Projection for DP-RP

From Theorem 3.1 and Theorem 3.2, it is clear that the noise magnitude of Gaussian noise in DP-RP directly depends on the $l_2$-sensitivity $\triangle_2$, which, according to (3), equals the largest row norm of the projection matrix $W$. Among the above mentioned distributions, the dense Rademacher projection ($s = 1$ in (7)) has $\triangle_2 = \frac{1}{\sqrt{k}}\beta \times \sqrt{k} = \beta$ which is independent of $p$. This could be much smaller than the dense Gaussian projection (i.e., DP-RP-G-OPT).

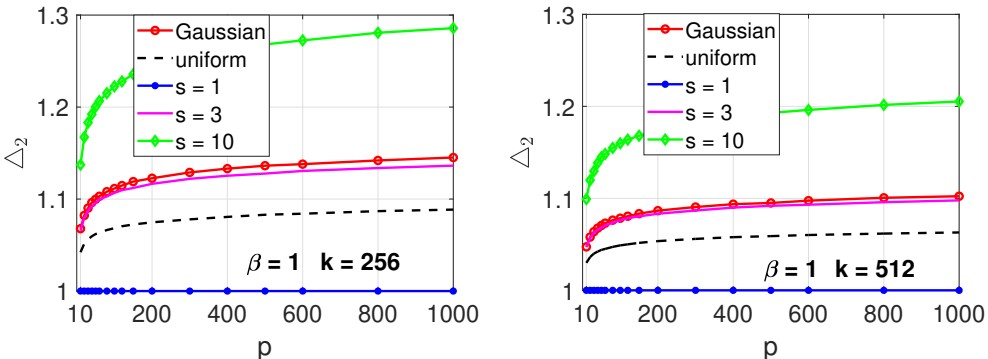

Figure 5: The $l_2$-sensitivity $\triangle_2$ (3) for different types of random projection matrices against the data dimensionality $p$, at $k = 256$ and $k = 512$, respectively. $\beta = 1$.

In Figure 5, we numerically simulate the $\triangle_2$ of different projection matrices, which shows that the Rademacher projection produces the smallest sensitivity. This, when plugged into the optimal Gaussian mechanism (Theorem 3.2), leads to smaller Gaussian noise variance needed.

## B.2    Rademacher Projection for DP-SignRP

From our analysis, it is clear that the flipping probability of DP-SignRP (both DP-SignRP-RR and DP-SignRP-RR-smooth) essentially depends on how concentrated the projected data is around zero. Particularly, $N_+$ in Algorithm 3, as given in Proposition 4.3, is a high probability upper bound on a Binomial random variable with success probability $Pr(\beta \max_{i=1,...,p} |w_i| \geq |w^T u|)$ with $w \sim N(0,1)$. In Algorithm 4, $L_j = \lceil \frac{|w_j^T u|}{\beta \max_{i=1,...,p} |W_{ij}|} \rceil$. For both quantities, a smaller value leads to a smaller sign flipping probability and thus better utility.

$N_+$ **in DP-SignRP-RR.** We first consider the $N_+$ in Algorithm 3, which determines the flipping probability $\frac{1}{e^{\epsilon/N_+}+1}$. Particularly, $N_+$ in Algorithm 3, as given in Proposition 4.3, is a high probability upper bound on a Binomial random variable with success probability

$$P_+ = Pr\left(\beta \max_{i=1,...,p} |w_i| \geq |w^T u|\right),  \tag{8}$$

where $w$ is the $p$-dimensional projection vector. When $w_i$ is sampled from the Rademacher distribution, i.e., $w_i \in \{-1, +1\}$ with equal probabilities, the probability calculation can be simplified:

$$P_{+,b} = Pr\left(\beta \max_{i=1,...,p} |w_i| \geq |\sum_{i=1}^p w_i u_i|\right) = Pr\left(\beta \geq |\sum_{i=1}^p w_i u_i|\right) \approx 2\Phi\left(\frac{\beta}{\|u\|}\right) - 1.  \tag{9}$$

Based on the central limit theorem, the normal approximation (9) is accurate unless $p$ is very small. Recall that, when $w_i$'s are sampled from the Gaussian distribution, we can calculate an upper bound in (20), which is re-written as below:

$$P_{+,g} = Pr\left(\beta \max_{i=1,...,p} |w_i| \geq |\sum_{i=1}^p w_i u_i|\right) \leq \int_0^\infty 2p[2\Phi(t)-1]^{p-1}[2\Phi(\beta t/\|u\|)-1]\phi(t)dt.  \tag{10}$$

Next, we provide a simulation study to justify the approximation and compare different distributions in terms of their impact on the probability (8), for $\beta = 1$ as well as $\beta = 0.1$. For simplicity, we simulate the data as a $p$-dimensional vector of uniform random numbers sampled from $unif[-1, 1]$. We experiment with five different choices of $w$: the standard Gaussian, the uniform, the "very sparse" distribution (7) with $s = 1$, $s = 3$, and $s = 10$. We vary $p$ from 10 to 1000. For each case, we repeat the simulations $10^7$ times to ensure sufficient accuracy. Figure 6 verifies that the two approximations (9) and (10) are accurate. In Figure 7, we provide the curves for more types of projection matrices. From both figures, we clearly see that using the Rademacher projection can considerably reduce (8) compared with Gaussian (and other) projections, leading to smaller $N_+$ value. This typically implies better utility.

$L_j$ **in DP-SignRP-RR-smooth.** Similarly, we numerically evaluate the $L_j$ in Algorithm 4. We run Algorithm 4 with $k = 512$, which gives 512 $L_j$ values. In Figure 8, we plot the proportion (or the approximated distribution) of the values of $L_j$ among $k$ projections. As we see, Rademacher projection produces least number of small $L_j$ values, and largest number of higher $L_j$ values. As the smooth flipping probability equals $\frac{1}{\exp(\frac{L_j}{k}\epsilon)+1}$, larger $L_j$ leads to smaller probability of sign flipping. Hence, Rademacher is again the best choice for the projection matrix.

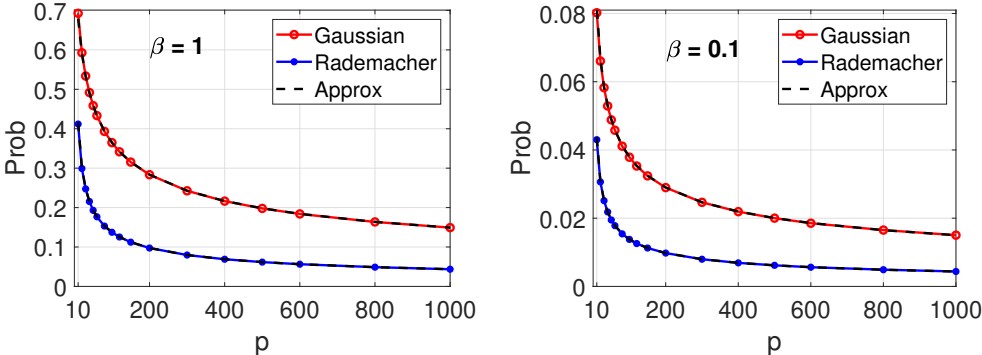

Figure 6: Simulations for evaluating (9) and (10), using two choices for $w$: the Gaussian distribution and the Rademacher distribution (i.e., (7) with $s = 1$). We plot the two upper bounds (9) and (10) as black dashed curves, which both overlap with their corresponding simulations.

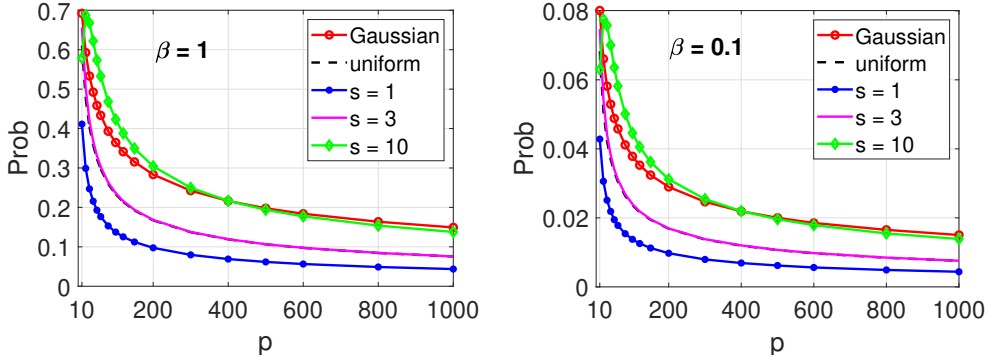

Figure 7: Simulations (same as in Figure 7) for evaluating (8), using five different choices for $w$: the Gaussian, the uniform, the "very sparse" distribution (7) with $s = 1, 3$ and 10. $s = 1$ is the Rademacher distribution. The data vector is simulated by sampling each entry from $unif[-1, 1]$.

## C    Comparison of DP-RP and DP-OPORP on Inner Product Estimation

In this section, we theoretically compare DP-RP and DP-OPORP. In Section B, we have shown that Rademacher projection requires lowest noise magnitude. Thus, we consider DP-RP with Rademacher projections here. For clarity, we summarize the algorithm in Algorithm 7. We name it "DP-RP-G-OPT-B", where "G" stands for the **G**aussian noise mechanism and "B" stands for "symmetric **B**ernoulli" projections.

---

**Algorithm 7:** DP-RP-G-OPT-B

---

1 **Input:** Data $u \in [-1, 1]^p$, privacy parameters $\epsilon > 0$, $\delta \in (0, 1)$, number of projections $k$
2 **Output:** $(\epsilon, \delta)$-differentially private random projections $\tilde{x} \in \mathbb{R}^k$
3 Apply RP $x = \frac{1}{\sqrt{k}} W^T u$, where $W \in \mathbb{R}^{p \times k}$ is a random Rademacher matrix
4 Generate iid random noise vector $G \in \mathbb{R}^k$ following $N(0, \sigma^2)$ where $\sigma$ is obtained by
   Theorem 3.2 with $\triangle_2 = \beta$
5 Return $\tilde{x} = x + G$

---

In our analysis, for simplicity we assume the data are normalized, i.e., the data vector has $l_2$ norm equal to 1. In this case, the inner product is also the cosine. The baseline method is the most straightforward: we add optimal Gaussian noise to each dimension of the original data (Raw-data-G-OPT). For this strategy, the sensitivity is also $\triangle_2 = \beta$. This means, when we compare all three methods: Raw-data-G-OPT, DP-RP-G-OPT-B, and DP-OPORP, the noise level $\sigma$ is the same. This makes it convenience to conduct the comparisons, from which we can gain valuable insights.

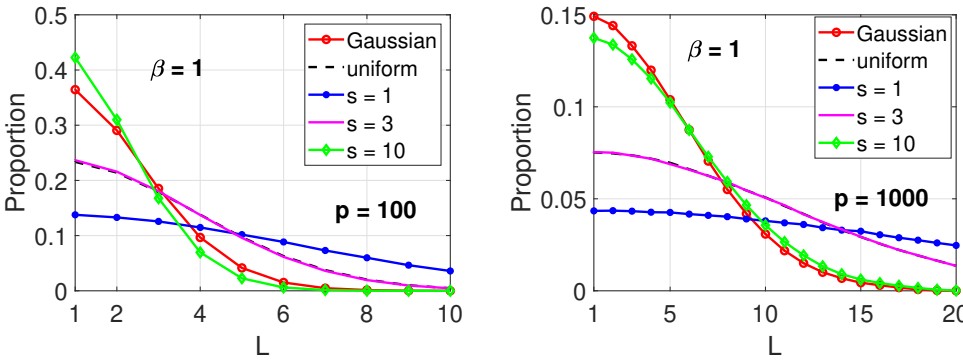

Figure 8: Simulations for evaluating $L_j$ in Algorithm 4, using different choices for $w$. $s = 1$ is the Rademacher distribution. Left: $p = 100$, right: $p = 1000$. The $y$-axis is the proportion (normalized histogram) of the values of all the $L_j$, $j = 1, ..., k$ computed using $k = 512$ projected samples.

**Theorem C.1** (Raw-data-G-OPT, i.e., adding optimal Gaussian noise on raw data). *Let $\sigma$ be the solution to (4) with $\triangle_2 = \beta$. For any $u, v \in \mathcal{U}$, let $\tilde{u}_i = u_i + a_i$ and $\tilde{v}_i = v_i + b_i$ be the DP noisy vectors, with $a_i, b_i \sim N(0, \sigma^2)$ i.i.d. Then, denote $\hat{g}_{org} = \sum_{i=1}^{p} \tilde{u}_i \tilde{v}_i$. we have*

$$\mathbb{E}[\hat{g}_{org}] = \sum_{i=1}^{p} u_i v_i, \qquad Var\left(\hat{g}_{org}\right) = \sigma^2 \sum_{i=1}^{p} \left(u_i^2 + v_i^2\right) + p\sigma^4. \qquad (11)$$

*Proof.* To add Gaussian noise to the original data, it suffices to find the sensitivity, which, by Definition 2.2, is $\triangle_2 = \beta$. Thus, the approach is $(\epsilon, \delta)$-DP according to the optimal Gaussian mechanism (Theorem 3.2). To compute the mean and variance, consider some $i \in [p]$. We have

$$\mathbb{E}\left[(u_i + a_i)(v_i + b_i)\right] = \mathbb{E}[u_i v_i + a_i v_i + b_i u_i + a_i b_i] = u_i v_i.$$

Thus, taking the sum implies $\mathbb{E}[\hat{g}_{org}] = \sum_{i=1}^{p} u_i v_i$. For the variance,

$$\mathbb{E}\left[(u_i + a_i)(v_i + b_i)\right]^2 = \mathbb{E}[u_i v_i + a_i v_i + b_i u_i + a_i b_i]^2 = u_i^2 v_i^2 + \sigma^2 \left(u_i^2 + v_i^2\right) + \sigma^4,$$

which leads to

$$Var\left((u_i + a_i)(v_i + b_i)\right) = \sigma^2 \left(u_i^2 + v_i^2\right) + \sigma^4.$$

Therefore, by independence,

$$Var\left(\hat{g}_{org}\right) = Var\left(\sum_{i=1}^{p} (u_i + a_i)(v_i + b_i)\right) = \sigma^2 \sum_{i=1}^{p} \left(u_i^2 + v_i^2\right) + p\sigma^4,$$

which proves the claim. $\qquad\qquad\qquad\qquad\qquad\qquad\qquad\qquad\qquad\qquad\qquad\qquad\qquad\square$

For DP-RP-G-OPT-B and DP-OPORP, we have the following results.

**Theorem C.2** (DP-RP-G-OPT-B inner product estimation). *Let $\sigma$ be the solution to (4) with $\triangle_2 = \beta$. In Algorithm 7, let $W \in \{-1, 1\}^{p \times k}$ be a Rademacher random matrix. Denote $x = \frac{1}{\sqrt{k}} W^T u$, $y = \frac{1}{\sqrt{k}} W^T v$, and $a$, $b$ are two random Gaussian noise vectors following $N(0, \sigma^2)$. Let $\hat{g}_{rp} = \sum_{j=1}^{k} (x_j + a_j)(y_j + b_j)$. Then, $\mathbb{E}[\hat{g}_{rp}] = \sum_{i=1}^{p} u_i v_i$, and*

$$Var\left(\hat{g}_{rp}\right) = \sigma^2 \sum_{i=1}^{p} \left(u_i^2 + v_i^2\right) + k\sigma^4 + \frac{1}{k} \left(\sum_{i=1}^{p} u_i^2 \sum_{i=1}^{p} v_i^2 + \left(\sum_{i=1}^{p} u_i v_i\right)^2 - 2\sum_{i=1}^{p} u_i^2 v_i^2\right). \qquad (12)$$

*Proof.* The conditional mean and variance can be computed as

$$\mathbb{E}\left[\sum_{j=1}^{k} (x_j + a_j)(y_j + b_j) \,\middle|\, x_j, y_j, j = 1, ..., k\right] = \sum_{j=1}^{k} x_j y_j,$$

$$Var\left(\sum_{j=1}^{k}(x_j+a_j)(y_j+b_j)\,|x_j,y_j,j=1,...,k\right)=\sigma^2\sum_{j=1}^{k}(x_j^2+y_j^2)+k\sigma^4,$$

where the variance calculation follows from Theorem C.1. Hence, we have

$$\mathbb{E}\left[\sum_{j=1}^{k}(x_j+a_j)(y_j+b_j)\right]=\mathbb{E}\left[\sum_{j=1}^{k}x_jy_j\right]=\sum_{i=1}^{p}u_iv_i,$$

$$Var\left(\hat{g}_{rp}\right)=\mathbb{E}\left[\sigma^2\sum_{j=1}^{k}(x_j^2+y_j^2)+k\sigma^4\right]+Var\left(\sum_{j=1}^{k}x_jy_j\right)$$

$$=\sigma^2\sum_{i=1}^{p}\left(u_i^2+v_i^2\right)+k\sigma^4+\frac{1}{k}\left(\sum_{i=1}^{p}u_i^2\sum_{i=1}^{p}v_i^2+\left(\sum_{i=1}^{p}u_iv_i\right)^2-2\sum_{i=1}^{p}u_i^2v_i^2\right). \quad (13)$$

In the above calculation, the formula of $Var\left(\sum_{j=1}^{k}x_jy_j\right)$ is from the result in [56] with $s=1$ for Rademacher distribution. $\qquad\square$

---

**Algorithm 8:** DP-OPORP

1 **Input:** Data $u\in[-1,1]^p$, privacy parameters $\epsilon>0$, $\delta\in(0,1)$, number of projections $k$
2 **Output:** Differentially private OPORP
3 Apply Algorithm 2 with a random Rademacher projection vector to obtain the OPORP $x$
4 Set sensitivity $\Delta_2=\beta$
5 Generate iid random vector $G\in\mathbb{R}^k$ following $N(0,\sigma^2)$ where $\sigma$ is computed by Theorem 3.2
6 Return $\tilde{x}=x+G$

---

**Theorem C.3** (DP-OPORP inner product estimation). *Let $\sigma$ be the solution to (4) with $\triangle_2=\beta$. Let $w\in\{-1,1\}^p$ be a Rademacher random vector. In Algorithm 8, let $x$ and $y$ be the OPORP of $u$ and $v$, and $a$, $b$ be two random Gaussian noise vectors following $N(0,\sigma^2)$. Denote $\hat{g}_{oporp}=\sum_{j=1}^{k}(x_j+a_j)(y_j+b_j)$. Then, $\mathbb{E}[\hat{g}_{oporp}]=\sum_{i=1}^{p}u_iv_i$, and*

$$Var\left(\hat{g}_{oporp}\right)=\sigma^2\sum_{i=1}^{p}\left(u_i^2+v_i^2\right)+k\sigma^4+\frac{1}{k}\left(\sum_{i=1}^{p}u_i^2\sum_{i=1}^{p}v_i^2+\left(\sum_{i=1}^{p}u_iv_i\right)^2-2\sum_{i=1}^{p}u_i^2v_i^2\right)\frac{p-k}{p-1}.$$
$$(14)$$

*Proof.* The proof is similar to that of Theorem C.2, with the help of the result in [57]. $\qquad\square$

The variance reduction factor $\frac{p-k}{p-1}$ can be quite beneficial when $p$ is not very large. Also, see [57] for the normalized estimators for both OPORP and VSRP (very sparse random projections). The normalization steps can substantially reduce the estimation variance.

**Comparison.** For the convenience of comparison, let us assume that the data are row-normalized, i.e., $\|u\|^2=1$ for all $u\in\mathcal{U}$. Let $\rho=\sum_{i=1}^{p}u_iv_i$. We have

$$Var\left(\hat{g}_{org}\right)=2\sigma^2+p\sigma^4,$$

$$Var\left(\hat{g}_{rp}\right)=2\sigma^2+k\sigma^4+\frac{1}{k}\left(1+\rho^2-2\sum_{i=1}^{p}u_i^2v_i^2\right),$$

$$Var\left(\hat{g}_{oporp}\right)=2\sigma^2+k\sigma^4+\frac{1}{k}\left(1+\rho^2-2\sum_{i=1}^{p}u_i^2v_i^2\right)\frac{p-k}{p-1}.$$

For high-dimensional data (large $p$), we see that $\hat{g}_{rp}$ and $\hat{g}_{oporp}$ has roughly the same variance, approximately $2\sigma^2 + k\sigma^4 + \frac{1}{k}$. We would like to compare this with $Var(\hat{g}_{org}) = 2\sigma^2 + p\sigma^4$ the variance for adding noise directly to the original data.

Let's define the ratio of the variances:

$$R = \frac{2\sigma^2 + p\sigma^4}{2\sigma^2 + k\sigma^4 + \frac{1}{k}} \sim \frac{p\sigma^4}{k\sigma^4} = \frac{p}{k} \quad \text{(if $p$ is large or $\sigma$ is high)} \tag{15}$$

to illustrate the benefit of RP-type algorithms (DP-RP and DP-OPORP) in protecting the privacy of the (high-dimensional) data. If $\frac{p}{k} = 100$, then it is possible that the ratio of the variances can be roughly 100. This would be a huge advantage. Figure 9 plots the ratio $R$ for $p = 1000$ and $p = 10000$ as well as a series of $k/p$ values, with respect to $\sigma$.

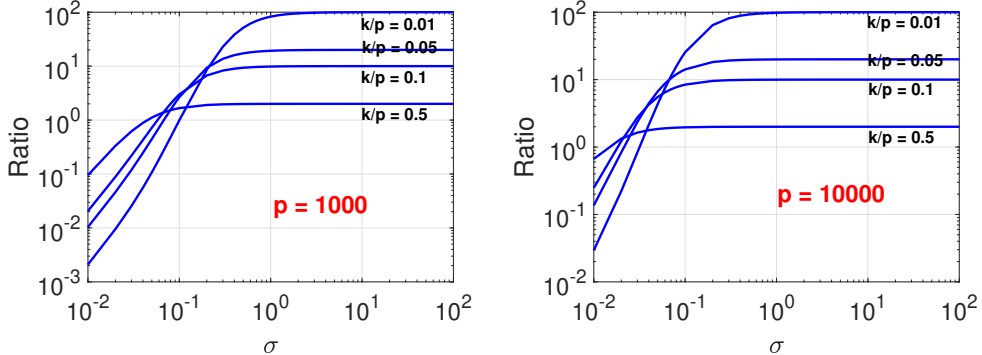

Figure 9: We plot the ratio of variances in (15) for $p = 1000$ and $p = 10000$. We choose $k$ values with $k/p \in \{0.01, 0.05, 0.1, 0.5\}$. Then for any $\sigma$ value, we are able to compute the ratio $R$. For larger $\sigma$, we have $R \sim \frac{p}{k}$ as expected. See Figure 10 for the relationship among $\sigma$, $\Delta$, and $\epsilon$ (and $\delta$).

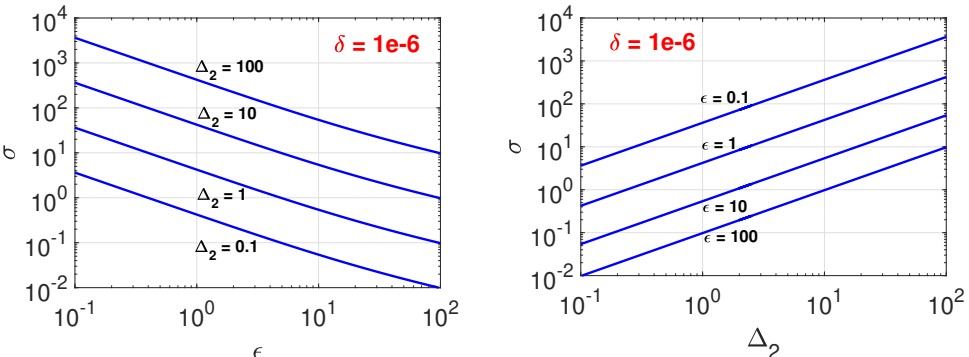

Figure 10: Left panel: the optimal Gaussian noise $\sigma$ versus $\epsilon$ for a series of $\Delta_2$ values, by solving the nonlinear equation (4) in Theorem 3.2, for $\delta = 10^{-6}$. Right panel: the optimal Gaussian noise $\sigma$ versus $\Delta_2$ for a series of $\epsilon$ values.

Figure 9 also illustrates when it might be a good strategy to directly add noise to the original data. For example, when $p = 1000$, the ratio can be below 1 if $\sigma < 0.1$. One can numerically verify that, (in Figure 10) in order for $\sigma < 0.1$ at $\Delta_2 = \beta = 1$, we need $\epsilon > 100$. In other words, adding noise to the raw data might be plausible when $\epsilon > 100$. In the literature, however, many DP applications typically require a much smaller $\epsilon$, such as $\epsilon \in [0.1, 20]$ (e.g., [43, 48]). Therefore, DP-RP and DP-OPORP is much better (i.e., has much smaller inner product estimation variance) than adding noise to the raw data in common privacy regimes.

# D   More Experiment Results

We provide the complete set of plots of our experimental results. In Figure 11 and Figure 12, we report the precision@10 and recall@100 curves of DP-RP variants and DP-OPORP on MNIST and CIFAR, respectively. In Figure 13, we report the test accuracy on the Webspam dataset of these methods. From all plots, we see that DP-RP-G-OPT-B and DP-OPORP perform equally the best on all the tasks, significantly better than the strategy of adding Gaussian noise to the raw data.

In Figure 14, we report the recall@100 metric of DP-SignOPORP methods in addition to the precision@10 metric shown in the main paper. For completeness, we also include the SVM test accuracy in Figure 15, which is the same as Figure 4 in the main context.

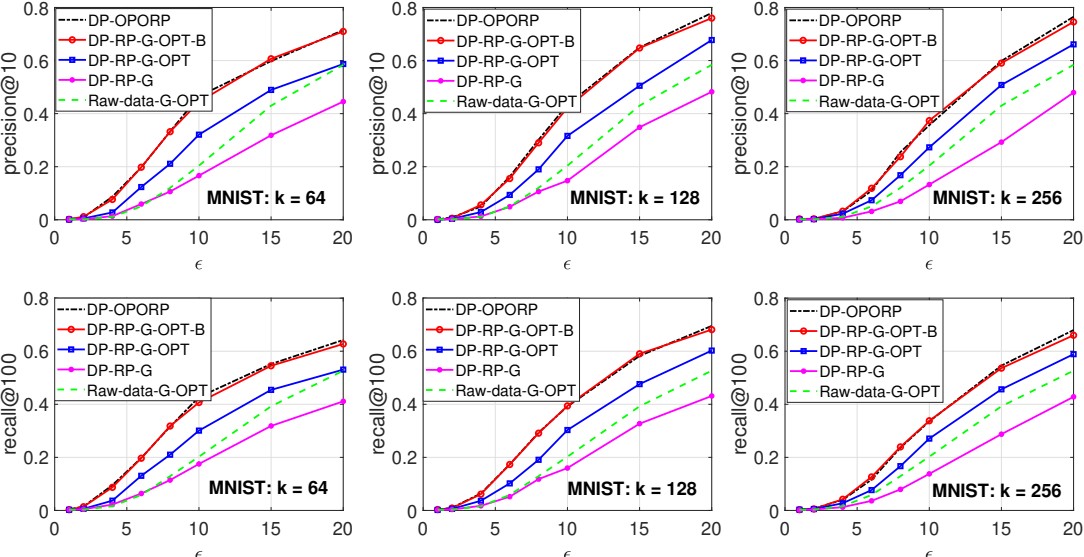

Figure 11: Retrieval recall and precision on MNIST, $\beta = 1$, $\delta = 10^{-6}$.

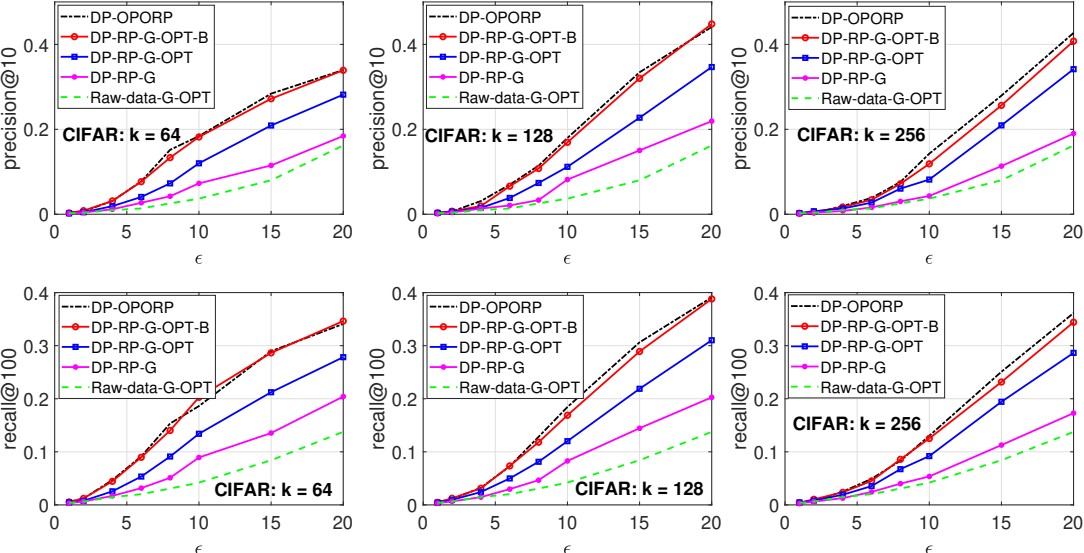

Figure 12: Retrieval recall and precision on CIFAR, $\beta = 1$, $\delta = 10^{-6}$.

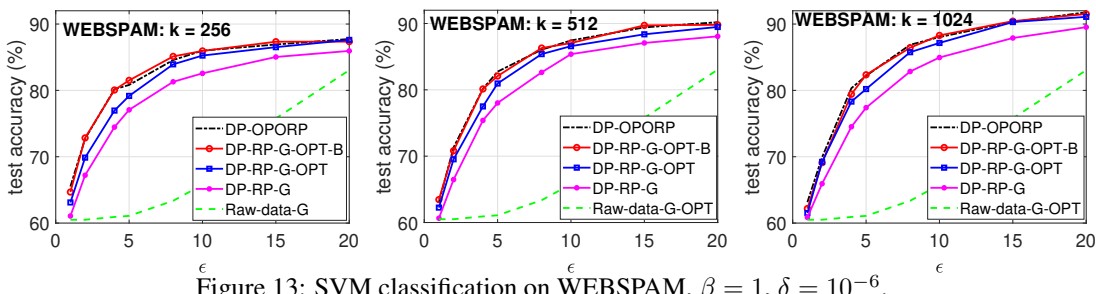

Figure 13: SVM classification on WEBSPAM, $\beta = 1$, $\delta = 10^{-6}$.

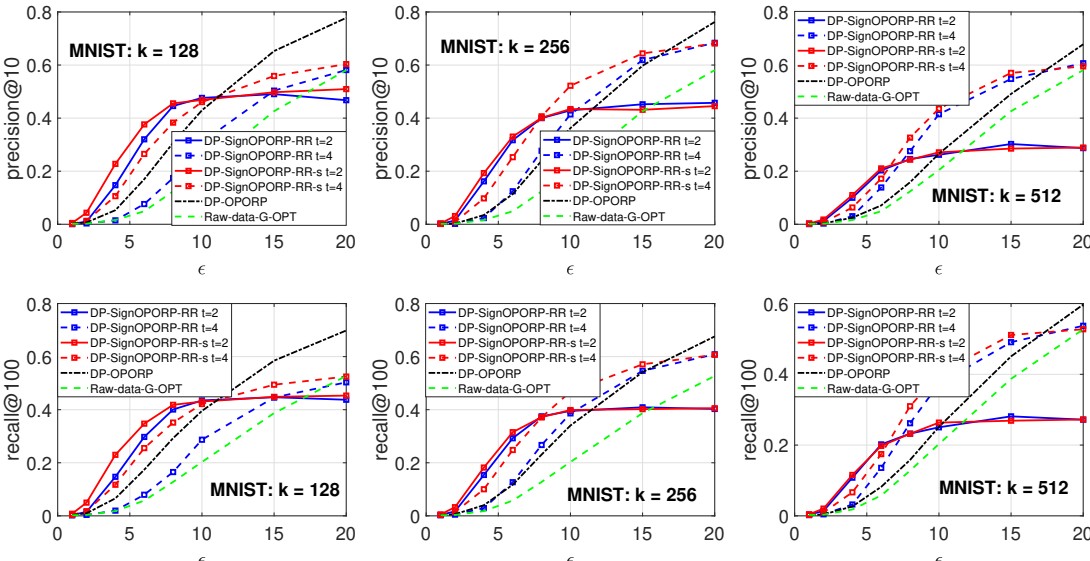

Figure 14: Retrieval on MNIST with DP-SignOPORP-RR and DP-SignOPORP-RR-smooth (in the caption, "-s" stands for "-smooth". For DP-OPORP and Raw-data-G-OPT, we let $\delta = 10^{-6}$.

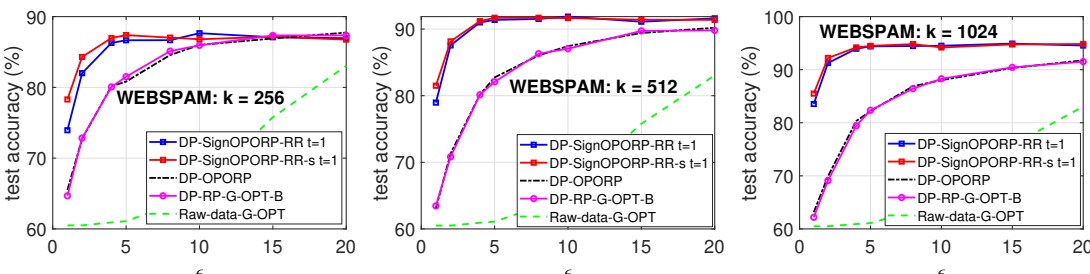

Figure 15: SVM classification on Webspam with DP-SignOPORP-RR and DP-SignOPORP-RR-s. For DP-OPORP and Raw-data-G-OPT, we let $\delta = 10^{-6}$.

# E  Deferred Proofs

The following lemma on Gaussian random variables will be used in our proof for Lemma 4.2, and may also be of independent interest.

**Lemma E.1.** *Let* $\begin{pmatrix} X \\ Y \end{pmatrix} \sim N \begin{pmatrix} \sigma_x^2 & \rho\sigma_x\sigma_y \\ \rho\sigma_x\sigma_y & \sigma_y^2 \end{pmatrix}$. *Denote* $r = \sigma_x/\sigma_y$. *Then we have:*

*1.* $Pr(|X| > |Y|) = \frac{1}{\pi} \left[ \tan^{-1} \left( \frac{r-\rho}{\sqrt{1-\rho^2}} \right) + \tan^{-1} \left( \frac{r+\rho}{\sqrt{1-\rho^2}} \right) \right]$. *When* $r \leq 1$, *the maximum is achieved at* $\rho = 0$, *i.e.,* $\max_\rho Pr(|X| < |Y|) = \frac{2}{\pi} \tan^{-1}(r)$.

*2. The conditional expectation:*

$$\mathbb{E}\left[|X|\,\big|\,|X| > |Y|\right] = \sigma_x \sqrt{\frac{\pi}{2}} \cdot \frac{\frac{r-\rho}{\sqrt{1+r^2-2r\rho}} + \frac{r+\rho}{\sqrt{1+r^2+2r\rho}}}{\tan^{-1}\left(\frac{r-\rho}{\sqrt{1-\rho^2}}\right) + \tan^{-1}\left(\frac{r+\rho}{\sqrt{1-\rho^2}}\right)}.$$

*3. The conditional tail probability: for any* $r > 0$, $\rho \in (-1, 1)$, *for any* $t > 0$,

$$Pr(|X| > t\,\big|\,|X| > |Y|) \leq \exp\left(-\frac{t^2}{2\sigma_x^2}\right).$$

*Proof.* The bivariate normal density function is

$$f(x, y) = \frac{1}{2\pi\sigma_x\sigma_y\sqrt{1-\rho^2}} \exp\left(-\frac{\frac{x^2}{\sigma_x^2} - \frac{2\rho xy}{\sigma_x\sigma_y} + \frac{y^2}{\sigma_y^2}}{2(1-\rho^2)}\right)$$

$$= \frac{1}{2\pi\sigma_x\sigma_y\sqrt{1-\rho^2}} \exp\left(-\frac{x^2}{2\sigma_x^2}\right) \exp\left(-\frac{(\frac{y}{\sigma_y} - \rho\frac{x}{\sigma_x})^2}{2(1-\rho^2)}\right).$$

Therefore, we have $\mathbb{E}\left[|X|\,\big|\,|X| > |Y|\right] = \frac{A}{P}$, with

$$A = \int_{-\infty}^{\infty} \frac{|x|}{\sqrt{2\pi}\sigma_x\sqrt{1-\rho^2}} \exp\left(-\frac{x^2}{2\sigma_x^2}\right) dx \int_{-|x|}^{|x|} \frac{1}{\sqrt{2\pi}\sigma_y} \exp\left(-\frac{(\frac{y}{\sigma_y} - \rho\frac{x}{\sigma_x})^2}{2(1-\rho^2)}\right) dy,$$

$$P = \int_{-\infty}^{\infty} \frac{1}{\sqrt{2\pi}\sigma_x\sqrt{1-\rho^2}} \exp\left(-\frac{x^2}{2\sigma_x^2}\right) dx \int_{-|x|}^{|x|} \frac{1}{\sqrt{2\pi}\sigma_y} \exp\left(-\frac{(\frac{y}{\sigma_y} - \rho\frac{x}{\sigma_x})^2}{2(1-\rho^2)}\right) dy.$$

Note that $P = Pr(|X| > |Y|)$ in the first statement of the theorem. Our calculation will use the following two identities involving the Gaussian functions [67]:

$$\int_0^\infty \phi(ax)\Phi(bx)dx = \frac{1}{2\pi|a|}\left(\frac{\pi}{2} + \tan^{-1}\left(\frac{b}{|a|}\right)\right), \tag{16}$$

$$\int_0^\infty x\phi(ax)\Phi(bx)dx = \frac{1}{2\sqrt{2\pi}}\left(1 + \frac{b}{\sqrt{1+b^2}}\right), \tag{17}$$

where $\phi(x)$ and $\Phi(x)$ are the pdf and cdf of the standard Gaussian distribution.

With a proper change of random variables, we can compute $A$ as

$$A = \int_{-\infty}^{\infty} \frac{|x|}{\sqrt{2\pi}\sigma_x\sqrt{1-\rho^2}} \exp\left(-\frac{x^2}{2\sigma_x^2}\right) dx \int_{\frac{-|x|}{\sigma_y} - \rho\frac{x}{\sigma_x}}^{\frac{|x|}{\sigma_y} - \rho\frac{x}{\sigma_x}} \sqrt{1-\rho^2}\frac{1}{\sqrt{2\pi}}e^{-s^2}ds$$

$$= \int_{-\infty}^{\infty} \frac{|x|}{\sqrt{2\pi}\sigma_x} \exp\left(-\frac{x^2}{2\sigma_x^2}\right) \left[\Phi\left(\frac{\frac{|x|}{\sigma_y} - \rho\frac{x}{\sigma_x}}{\sqrt{1-\rho^2}}\right) - \Phi\left(\frac{\frac{-|x|}{\sigma_y} - \rho\frac{x}{\sigma_x}}{\sqrt{1-\rho^2}}\right)\right] dx$$

$$= \int_{-\infty}^{\infty} \frac{\sigma_x|t|}{\sqrt{2\pi}} \exp\left(-\frac{t^2}{2}\right) \left[\Phi\left(\frac{\frac{\sigma_x}{\sigma_y}|t| - \rho t}{\sqrt{1-\rho^2}}\right) - \Phi\left(-\frac{\frac{\sigma_x}{\sigma_y}|t| - \rho t}{\sqrt{1-\rho^2}}\right)\right] dt$$

$$:= A_1 - A_2.$$

For the first term we have

$$A_1 = \sigma_x \left[ \int_0^\infty \frac{t}{\sqrt{2\pi}} \exp\left(-\frac{t^2}{2}\right) \Phi\left(\frac{\frac{\sigma_x}{\sigma_y} - \rho}{\sqrt{1-\rho^2}} t\right) dt + \int_{-\infty}^0 \frac{-t}{\sqrt{2\pi}} \exp\left(-\frac{t^2}{2}\right) \Phi\left(-\frac{\frac{\sigma_x}{\sigma_y} + \rho}{\sqrt{1-\rho^2}} t\right) dt \right]$$

$$= \sigma_x \left[ \int_0^\infty \frac{t}{\sqrt{2\pi}} \exp\left(-\frac{t^2}{2}\right) \Phi\left(\frac{\frac{\sigma_x}{\sigma_y} - \rho}{\sqrt{1-\rho^2}} t\right) dt + \int_0^\infty \frac{s}{\sqrt{2\pi}} \exp\left(-\frac{s^2}{2}\right) \Phi\left(\frac{\frac{\sigma_x}{\sigma_y} + \rho}{\sqrt{1-\rho^2}} s\right) ds \right]$$

$$= \sigma_x \left[ \frac{1}{2\sqrt{2\pi}} \left( 1 + \frac{\frac{\frac{\sigma_x}{\sigma_y} - \rho}{\sqrt{1-\rho^2}}}{\sqrt{1 + \frac{(\frac{\sigma_x}{\sigma_y} - \rho)^2}{1-\rho^2}}} \right) + \frac{1}{2\sqrt{2\pi}} \left( 1 + \frac{\frac{\frac{\sigma_x}{\sigma_y} + \rho}{\sqrt{1-\rho^2}}}{\sqrt{1 + \frac{(\frac{\sigma_x}{\sigma_y} + \rho)^2}{1-\rho^2}}} \right) \right]$$

$$= \sigma_x \left[ \frac{1}{\sqrt{2\pi}} + \frac{1}{2\sqrt{2\pi}} \left( \frac{r - \rho}{\sqrt{1 + r^2 - 2r\rho}} + \frac{r + \rho}{\sqrt{1 + r^2 + 2r\rho}} \right) \right],$$

where we denote $r = \frac{\sigma_x}{\sigma_y}$ and use (17). Similarly, we have that

$$A_2 = \sigma_x \left[ \int_0^\infty \frac{t}{\sqrt{2\pi}} \exp\left(-\frac{t^2}{2}\right) \Phi\left(-\frac{r + \rho}{\sqrt{1-\rho^2}} t\right) dt + \int_{-\infty}^0 \frac{-t}{\sqrt{2\pi}} \exp\left(-\frac{t^2}{2}\right) \Phi\left(\frac{r - \rho}{\sqrt{1-\rho^2}} t\right) dt \right]$$

$$= \sigma_x \left[ \int_0^\infty \frac{t}{\sqrt{2\pi}} \exp\left(-\frac{t^2}{2}\right) \Phi\left(-\frac{r + \rho}{\sqrt{1-\rho^2}} t\right) dt + \int_0^\infty \frac{s}{\sqrt{2\pi}} \exp\left(-\frac{s^2}{2}\right) \Phi\left(-\frac{r - \rho}{\sqrt{1-\rho^2}} s\right) ds \right]$$

$$= \sigma_x \left[ \frac{1}{\sqrt{2\pi}} - \frac{1}{2\sqrt{2\pi}} \left( \frac{r - \rho}{\sqrt{1 + r^2 - 2r\rho}} + \frac{r + \rho}{\sqrt{1 + r^2 + 2r\rho}} \right) \right].$$

Therefore, we obtain

$$A(\rho, r) = A_1 - A_2 = \frac{\sigma_x}{\sqrt{2\pi}} \left( \frac{r - \rho}{\sqrt{1 + r^2 - 2r\rho}} + \frac{r + \rho}{\sqrt{1 + r^2 + 2r\rho}} \right). \tag{18}$$

To compute $P$, by doing a similar change of variables, we have

$$P = \int_{-\infty}^\infty \frac{1}{\sqrt{2\pi}} \exp\left(-\frac{t^2}{2}\right) \left[ \Phi\left(\frac{r|t| - \rho t}{\sqrt{1-\rho^2}}\right) - \Phi\left(-\frac{r|t| - \rho t}{\sqrt{1-\rho^2}}\right) \right] dt$$

$$:= P_1 - P_2.$$

Using (16), we obtain

$$P_1 = \int_0^\infty \frac{1}{\sqrt{2\pi}} \exp\left(-\frac{t^2}{2}\right) \Phi\left(\frac{r - \rho}{\sqrt{1-\rho^2}} t\right) dt + \int_{-\infty}^0 \frac{1}{\sqrt{2\pi}} \exp\left(-\frac{t^2}{2}\right) \Phi\left(-\frac{r + \rho}{\sqrt{1-\rho^2}} t\right) dt$$

$$= \int_0^\infty \frac{1}{\sqrt{2\pi}} \exp\left(-\frac{t^2}{2}\right) \Phi\left(\frac{r - \rho}{\sqrt{1-\rho^2}} t\right) dt + \int_0^\infty \frac{1}{\sqrt{2\pi}} \exp\left(-\frac{s^2}{2}\right) \Phi\left(\frac{r + \rho}{\sqrt{1-\rho^2}} s\right) ds$$

$$= \frac{1}{2\pi} \left( \frac{\pi}{2} + \tan^{-1}\left(\frac{r - \rho}{\sqrt{1-\rho^2}}\right) \right) + \frac{1}{2\pi} \left( \frac{\pi}{2} + \tan^{-1}\left(\frac{r + \rho}{\sqrt{1-\rho^2}}\right) \right)$$

$$= \frac{1}{2} + \frac{1}{2\pi} \left[ \tan^{-1}\left(\frac{r - \rho}{\sqrt{1-\rho^2}}\right) + \tan^{-1}\left(\frac{r + \rho}{\sqrt{1-\rho^2}}\right) \right],$$

$$P_2 = \frac{1}{2} - \frac{1}{2\pi} \left[ \tan^{-1}\left(\frac{r - \rho}{\sqrt{1-\rho^2}}\right) + \tan^{-1}\left(\frac{r + \rho}{\sqrt{1-\rho^2}}\right) \right],$$

which leads to

$$P(\rho, r) = P_1 - P_2 = \frac{1}{\pi} \left[ \tan^{-1}\left(\frac{r - \rho}{\sqrt{1-\rho^2}}\right) + \tan^{-1}\left(\frac{r + \rho}{\sqrt{1-\rho^2}}\right) \right] \tag{19}$$

Therefore, we know that

$$\mathbb{E}\left[|X|\,\big|\,|X|>|Y|\right] = \frac{A(\rho,r)}{P(\rho,r)} = \sigma_x\sqrt{\frac{\pi}{2}}\cdot\frac{\frac{r-\rho}{\sqrt{1+r^2-2r\rho}}+\frac{r+\rho}{\sqrt{1+r^2+2r\rho}}}{\tan^{-1}\left(\frac{r-\rho}{\sqrt{1-\rho^2}}\right)+\tan^{-1}\left(\frac{r+\rho}{\sqrt{1-\rho^2}}\right)},$$

with $r=\sigma_x/\sigma_y$. We now investigate the derivative of $P$. By some algebra, we can show that

$$\frac{\partial P(\rho,r)}{\partial\rho} = \frac{2r\rho(r^2-1)}{(1+r^2-2r\rho)(1+r^2+2r\rho)\sqrt{1-\rho^2}}.$$

When $0<r\leq 1$, $\frac{\partial P(\rho,r)}{\partial\rho}\geq 0$ when $\rho\leq 0$ and $\frac{\partial P(\rho,r)}{\partial\rho}\leq 0$ when $\rho>0$. Therefore, $\max_\rho P(\rho,r)=P(0,r)=\frac{2}{\pi}\tan^{-1}(r)$.

**Tail bound.** By our previous calculations, the conditional distribution of $X$ given $|X|>|Y|$ is

$$f(x\big||X|>|Y|) = \frac{\frac{1}{\sqrt{2\pi}}\exp\left(-\frac{x^2}{2}\right)\left[\Phi\left(\frac{r|x|-\rho x}{\sqrt{1-\rho^2}}\right)-\Phi\left(-\frac{r|x|-\rho x}{\sqrt{1-\rho^2}}\right)\right]}{P},\quad x\in\mathbb{R},$$

with $P=Pr(|X|>|Y|)$ in (19) the normalizing constant to make the integral equal to 1.

The conditional tail probability can be computed as follows. For some $t>0$, by symmetry,

$$Pr(|X|>t,|X|>|Y|)$$
$$=2\int_t^\infty\frac{1}{\sqrt{2\pi}\sigma_x}\exp\left(-\frac{x^2}{2\sigma_x^2}\right)\left[\Phi\left(\frac{r|x|-\rho x}{\sqrt{1-\rho^2}}\right)-\Phi\left(-\frac{r|x|-\rho x}{\sqrt{1-\rho^2}}\right)\right]dx$$
$$=2\int_{\frac{t}{\sigma_x}}^\infty\frac{1}{\sqrt{2\pi}}\exp\left(-\frac{x^2}{2}\right)\left[\Phi\left(\frac{r|x|-\rho x}{\sqrt{1-\rho^2}}\right)-\Phi\left(-\frac{r|x|-\rho x}{\sqrt{1-\rho^2}}\right)\right]dx$$
$$:=2(\tilde{P}_1-\tilde{P}_2).$$

For $\tilde{P}_1$, using polar coordinates we have

$$\tilde{P}_1 = \frac{1}{2\pi}\int_{\frac{t}{\sigma_x}}^\infty e^{-\frac{x^2}{2}}dx\int_{-\infty}^{\frac{r-\rho}{\sqrt{1-\rho^2}}x}e^{-\frac{y^2}{2}}dy$$
$$=\frac{1}{2\pi}\int_{-\frac{\pi}{2}}^{\tan^{-1}\left(\frac{r-\rho}{\sqrt{1-\rho^2}}\right)}d\theta\int_{\frac{t}{\sigma_x\cos(\theta)}}^\infty e^{-\frac{r^2}{2}}r\,dr$$
$$=\frac{1}{2\pi}\int_{-\frac{\pi}{2}}^{\tan^{-1}\left(\frac{r-\rho}{\sqrt{1-\rho^2}}\right)}\exp\left(-\frac{t^2}{2\sigma_x^2\cos^2(\theta)}\right)d\theta.$$

Similarly,

$$\tilde{P}_2 = \frac{1}{2\pi}\int_{-\frac{\pi}{2}}^{\tan^{-1}\left(-\frac{r+\rho}{\sqrt{1-\rho^2}}\right)}\exp\left(-\frac{t^2}{2\sigma_x^2\cos^2(\theta)}\right)d\theta.$$

Therefore, we obtain

$$Pr(|X|>t,|X|>|Y|) = \frac{1}{\pi}\int_{\tan^{-1}\left(-\frac{r+\rho}{\sqrt{1-\rho^2}}\right)}^{\tan^{-1}\left(\frac{r-\rho}{\sqrt{1-\rho^2}}\right)}\exp\left(-\frac{t^2}{2\sigma_x^2\cos^2(\theta)}\right)d\theta$$
$$=\frac{1}{\pi}\int_{-\tan^{-1}\left(\frac{r+\rho}{\sqrt{1-\rho^2}}\right)}^{\tan^{-1}\left(\frac{r-\rho}{\sqrt{1-\rho^2}}\right)}\exp\left(-\frac{t^2}{2\sigma_x^2\cos^2(\theta)}\right)d\theta$$
$$\leq e^{-\frac{t^2}{2\sigma_x^2}}\frac{1}{\pi}\int_{-\tan^{-1}\left(\frac{r+\rho}{\sqrt{1-\rho^2}}\right)}^{\tan^{-1}\left(\frac{r-\rho}{\sqrt{1-\rho^2}}\right)}d\theta,$$

since $\cos^2(\theta) \in [0,1]$. Notice that $P$ in (19) can be written as $P = \frac{1}{\pi} \int_{-\tan^{-1}(\frac{r+\rho}{\sqrt{1-\rho^2}})}^{\tan^{-1}(\frac{r-\rho}{\sqrt{1-\rho^2}})} d\theta$. Hence, we know that the conditional tail probability is

$$Pr(|X| > t \big| |X| > |Y|) = \frac{Pr(|X| > t, |X| > |Y|)}{Pr(|X| > |Y|)}$$

$$\leq \exp\left(-\frac{t^2}{2\sigma_x^2}\right), \quad \forall r > 0, \rho \in (-1,1).$$

At the boundaries $\rho = 1$, $\rho = -1$, one can verify $Pr(|X| > |Y|) = 0$. This concludes the proof. $\square$

## E.1 Proof of Lemma 4.2

*Proof.* By Lemma E.1 and independence, we know that $Pr(\max_{i=1,\dots,p} |X| > |Y|)$ reaches its maximum when $X_i$ is independent of $Y$, i.e., when $\rho(X_i, Y) = 0, \forall i = 1, \dots, p$. Since $|X_i|$ follows a half-normal distribution with cdf being $\text{erf}(\frac{x}{\sqrt{2}\sigma_x})$, we have

$$Pr(\max_{i=1,\dots,p} |X_i| \leq t) = \text{erf}\left(\frac{t}{\sqrt{2}\sigma_x}\right)^p = \left[2\Phi\left(\frac{x}{\sigma_x}\right) - 1\right]^p,$$

and probability density function $g(x) = 2p[\Phi(\frac{x}{\sigma_x}) - 1]^{p-1} \frac{1}{\sqrt{2\pi}\sigma_x} e^{-\frac{x^2}{2\sigma_x^2}}$. When $Y$ is independent of all $X_i$'s (which gives the upper bound), we have

$$Pr(\max_{i=1,\dots,p} |X| > |Y|) = \int_0^\infty 2p\left[\Phi\left(\frac{x}{\sigma_x}\right) - 1\right]^{p-1} \frac{1}{\sqrt{2\pi}\sigma_x} e^{-\frac{x^2}{2\sigma_x^2}} Pr(|Y| < x) dx$$

$$= \int_0^\infty 2p\left[\Phi\left(\frac{x}{\sigma_x}\right) - 1\right]^{p-1} \frac{1}{\sqrt{2\pi}\sigma_x} e^{-\frac{x^2}{2\sigma_x^2}} \text{erf}\left(\frac{x}{\sqrt{2}\sigma_y}\right) dx$$

$$= \int_0^\infty 2p[2\Phi(t) - 1]^{p-1} [2\Phi(rt) - 1]\phi(t) dt,$$

with a proper change of variables. This gives an upper bound as shown above. $\square$

## E.2 Proof of Proposition 4.3

*Proof.* Consider a single Gaussian projection vector $w$ with iid $N(0,1)$ entries. Since $w^T u = \sum_{i=1}^p u_i w_i$ and each $w_i \sim N(0,1)$, we know that $\begin{pmatrix} \beta w_i \\ x \end{pmatrix} \sim N\begin{pmatrix} \beta^2 & \rho_i\beta\|u\| \\ \rho_i\beta\|u\| & \|u\|^2 \end{pmatrix}$ where $\rho_i = \frac{u_i}{\|u\|}$ is the correlation coefficient. Since $|w^T(u - u')| \leq \beta \max_{i=1,\dots,p} |w_i|$ by Definition 2.2 of $\beta$-neighboring (and more generally, when $\|u - u'\|_1 \leq \beta$), we have

$$Pr(\max_{u' \in Nb(u)} |w^T(u - u')| \geq |w^T u|) = Pr(\beta \max_{i=1,\dots,p} |w_i| \geq |w^T u|).$$

Note that, $\beta \max_{i=1,\dots,p} |w_i| \geq |w^T u|$ is a necessary condition for the event that there exists a neighbor such that $sign(w^T u) \neq sign(w^T u')$. Denote $I = \mathbb{1}\{\beta \max_{i=1,\dots,p} |w_i| \geq |w^T u|\}$. Applying Lemma 4.2 with $r = \beta/\|u\| \leq 1$ yields

$$\mathbb{E}[I] = Pr(\beta \max_{i=1,\dots,p} |w_i| \geq |w^T u|)$$

$$\leq F_{\|u\|,p} = \int_0^\infty 2p[2\Phi(t) - 1]^{p-1} [2\Phi(rt) - 1]\phi(t) dt \qquad (20)$$

as given by (5). Let $I_j$ be the corresponding indicator function w.r.t. each column in the projection matrix $W$. Denote $N_+ = \sum_{j=1}^k I_j$, and by the above reasoning, we know that $|S| \leq N_+$ where $S$ is defined in the theorem. Since the columns of $W$ are independent, $N_+$ follows a $Binomial(k, \mathbb{E}[I])$ distribution with $k$ trials and success probability $\mathbb{E}[I]$ bounded as above. Therefore, with probability $1 - \delta$, we have $N_+ \leq B^{-1}(\delta; F_{\|u\|,p}, k)$, where $B^{-1}$ is the inverse cdf of the Binomial distribution. This proves the claim as desired. $\square$

### E.3 Proof of Theorem 4.4

*Proof.* Let $s = sign(W^T u) \in \{-1, +1\}^k$, $s' = sign(W^T u') \in \{-1, +1\}^k$. We denote the collision probability of non-private SignRP as

$$P_{SRP} = Pr(s_{1j} = s_{2j}) = 1 - \frac{\cos^{-1}(\rho)}{\pi} = 1 - \frac{\theta}{\pi}.$$

Hence, the collision probability of DP-SignRP-RR can be computed as

$$\tilde{P} := Pr(\tilde{s}_{1j} = \tilde{s}_{2j}) = Pr(s_{1j} = s_{2j}, \text{both change sign or not change sign})$$
$$+ Pr(s_{1j} \neq s_{2j}, \text{one sign changes})$$
$$= P_{SRP}[(\frac{e^{\epsilon'}}{e^{\epsilon'} + 1})^2 + (\frac{1}{e^{\epsilon'} + 1})^2] + 2(1 - P_{SRP})\frac{e^{\epsilon'}}{(e^{\epsilon'} + 1)^2}$$
$$= P_{SRP}\frac{(e^{\epsilon'} - 1)^2}{(e^{\epsilon'} + 1)^2} + \frac{2e^{\epsilon'}}{(e^{\epsilon'} + 1)^2},$$

which increases linearly in $P_{SRP}$. Thus, it holds that

$$\mathbb{E}[\hat{P}_{RR}] = \frac{(e^{\epsilon'} + 1)^2}{(e^{\epsilon'} - 1)^2}\tilde{P} - \frac{2e^{\epsilon'}}{(e^{\epsilon'} - 1)^2} = P_{SRP} = 1 - \frac{\theta}{\pi},$$

which implies $\mathbb{E}[\hat{\theta}_{RR}] = \pi\left(1 - (1 - \frac{\theta}{\pi})\right) = \theta$. To compute the variance, we first estimate $\theta = \cos^{-1}(\rho)$ by

$$\hat{\theta} = \pi(1 - \hat{P}_{RR}).$$

Then according to the Central Limit Theorem (CLT), for the sample mean of iid Bernoulli's, as $k \to \infty$, we have

$$\frac{1}{k}\sum_{j=1}^{k} \mathbb{1}\{\tilde{s}_{1j} = \tilde{s}_{2j}\} \to N(\tilde{P}, \frac{\tilde{P}(1 - \tilde{P})}{k}).$$

As a result, we have $\hat{\theta} \to N(\theta, \frac{V_{RR}}{k})$, where

$$V_{RR} = \frac{\pi^2(e^{\epsilon'} + 1)^4}{(e^{\epsilon'} - 1)^4}\left[(1 - \frac{\theta}{\pi})\frac{(e^{\epsilon'} - 1)^2}{(e^{\epsilon'} + 1)^2} + \frac{2e^{\epsilon'}}{(e^{\epsilon'} + 1)^2}\right]\left[\frac{e^{2\epsilon'} + 1}{(e^{\epsilon'} + 1)^2} - (1 - \frac{\theta}{\pi})\frac{(e^{\epsilon'} - 1)^2}{(e^{\epsilon'} + 1)^2}\right]$$
$$= \frac{\pi^2(e^{\epsilon'} + 1)^4}{(e^{\epsilon'} - 1)^4}\left[(1 - \frac{\theta}{\pi})\frac{(e^{\epsilon'} - 1)^2}{(e^{\epsilon'} + 1)^2} + \frac{2e^{\epsilon'}}{(e^{\epsilon'} + 1)^2}\right]\left[\frac{\theta}{\pi}\frac{(e^{\epsilon'} - 1)^2}{(e^{\epsilon'} + 1)^2} + \frac{2e^{\epsilon'}}{(e^{\epsilon'} + 1)^2}\right]$$
$$= \frac{\pi^2\theta}{\pi}(1 - \frac{\theta}{\pi}) + (1 - \frac{\theta}{\pi})\frac{2e^{\epsilon'}}{(e^{\epsilon'} - 1)^2} + \frac{\theta}{\pi}\frac{2e^{\epsilon'}}{(e^{\epsilon'} - 1)^2} + \frac{4e^{2\epsilon'}}{(e^{\epsilon'} - 1)^4}$$
$$= \theta(\pi - \theta) + \frac{2\pi^2 e^{\epsilon'}}{(e^{\epsilon'} - 1)^2} + \frac{4\pi^2 e^{2\epsilon'}}{(e^{\epsilon'} - 1)^4}.$$

We conclude the proof by replacing $\epsilon' = \epsilon/N_+$. $\qquad\square$

### E.4 Proof of Theorem 4.5

*Proof.* Let us consider a single projection vector $w_j = W_{[:,j]}$. Denote $x_j = w_j^T u$ and $x'_j = w_j^T u'$ for a neighboring data $u'$ of $u$, and $s_j = sign(x_j)$, $s'_j = sign(x'_j)$. Also, let $L_j = \lceil \frac{|x_j|}{\beta \max_{i=1,...,p} |W_{ij}|} \rceil$ and $L'_j = \lceil \frac{|x'_j|}{\beta \max_{i=1,...,p} |W_{ij}|} \rceil$. W.l.o.g., we can assume $s_j = 1$ by the symmetry of random projection and the symmetry of DP. Consider two cases:

- Case I: $L_j \geq 2$. In this case, we know that $s'_j = s_j$, i.e., the change from $u$ to $u'$ will not change the sign of the projection. Thus, in Algorithm 4, we have

$$\frac{Pr(\tilde{s}_j = 1)}{Pr(\tilde{s}'_j = 1)} = \exp(\frac{L_j - L'_j}{k}\epsilon)\frac{\exp(\frac{L'_j}{k}\epsilon) + 1}{\exp(\frac{L_j}{k}\epsilon) + 1}.$$

By the definition of $\beta$-adjacency, $|L_j - L'_j|$ equals either 0 or 1. When $L_j = L'_j$, $\frac{Pr(\tilde{s}_j=1)}{Pr(\tilde{s}'_j=1)} = 1$. When $L_j - L'_j = 1$, we have

$$\frac{Pr(\tilde{s}_j = 1)}{Pr(\tilde{s}'_j = 1)} = \frac{\exp(\frac{L_j}{k}\epsilon) + \exp(\frac{1}{k}\epsilon)}{\exp(\frac{L_j}{k}\epsilon) + 1}.$$

Hence, we have $1 \leq \frac{Pr(\tilde{s}_j=1)}{Pr(\tilde{s}'_j=1)} \leq e^{\frac{\epsilon}{k}}$ by the numeric identity $1 \leq \frac{a+c}{b+c} \leq \frac{a}{b}$ for $a \geq b > 0$ and $c > 0$. Thus, by symmetry, $e^{-\frac{\epsilon}{k}} \leq \frac{Pr(\tilde{s}_j=1)}{Pr(\tilde{s}'_j=1)} \leq e^{\frac{\epsilon}{k}}$. On the other hand,

$$\frac{Pr(\tilde{s}_j = -1)}{Pr(\tilde{s}'_j = -1)} = \frac{\exp(\frac{L'_j}{k}\epsilon) + 1}{\exp(\frac{L_j}{k}\epsilon) + 1}.$$

Similarly, when $L_j = L'_j$, the ratio equals 1. When $L_j = L'_j - 1$, we have $\frac{Pr(\tilde{s}_j=-1)}{Pr(\tilde{s}'_j=-1)} \leq \exp(\frac{L'_j}{k}\epsilon - \frac{L_j}{k}\epsilon) = e^{\frac{\epsilon}{k}}$. By symmetry we obtain $e^{-\frac{\epsilon}{k}} \leq \frac{Pr(\tilde{s}_j=-1)}{Pr(\tilde{s}'_j=-1)} \leq e^{\frac{\epsilon}{k}}$.

- Case II: $L_j = 1$. In this case, $s_j$ might be different from $s'_j$. First, if $L'_j = 2$, then the above analysis also applies that $\frac{Pr(\tilde{s}_j=1)}{Pr(\tilde{s}'_j=1)}$ and $\frac{Pr(\tilde{s}_j=-1)}{Pr(\tilde{s}'_j=-1)}$ are both lower and upper bounded by $e^{-\frac{\epsilon}{k}}$ and $e^{\frac{\epsilon}{k}}$, respectively. It suffices to examine the case when $L'_j = 1$. In this case, if $s'_j = s_j = 1$ then the probability ratios simply equal 1. If $s'_j = -1$, we have

$$\frac{Pr(\tilde{s}_j = 1)}{Pr(\tilde{s}'_j = 1)} = \frac{\frac{\exp(\frac{\epsilon}{k})}{\exp(\frac{\epsilon}{k})+1}}{\frac{1}{\exp(\frac{\epsilon}{k})+1}} = e^{\frac{\epsilon}{k}}, \quad \frac{Pr(\tilde{s}_j = -1)}{Pr(\tilde{s}'_j = -1)} = \frac{\frac{1}{\exp(\frac{\epsilon}{k})+1}}{\frac{\exp(\frac{\epsilon}{k})}{\exp(\frac{\epsilon}{k})+1}} = e^{-\frac{\epsilon}{k}}.$$

Combining two cases, we have that $\log \frac{Pr(\tilde{s}_j=t)}{Pr(\tilde{s}'_j=t)} \leq \frac{\epsilon}{k}$, for $t = -1, 1$, and for all $j = 1, ..., k$. That is, each single perturbed sign achieves $\frac{\epsilon}{k}$-DP. Since the $k$ projections are independent, by Theorem 2.1, we know that the output bit vector $\tilde{s} = [\tilde{s}_1, ..., \tilde{s}_k]$ is $\epsilon$-DP as claimed. $\qquad\square$

