# Contents

# A iDP-SignRP Under Individual Differential Privacy (iDP)

## A.1 Relaxation: Individual Differential Privacy (iDP)

Many extensions or relaxation of DP have been proposed to improve the utility of DP mechanisms. Examples include Concentrated Differential Privacy [4], Rényi Differential Privacy [10], and Gaussian Differential Privacy [3]. These alternatives provide better composition properties than the composition theorems of DP, thus reducing the noise needed [6]. Another possible direction to elevate the empirical performance of DP is to relax the DP definition by constraining the scope of neighboring datasets depending on the specific use case of DP [12; 2]. In this paper, we consider the concept called "individual differential privacy" (iDP), also known as "data-centric DP", as follows.

**Definition A.1** (Individual DP [12]). *Given a dataset $U$, an algorithm $\mathcal{M}$ satisfies $(\epsilon, \delta)$-iDP for $U$ if for any dataset $U'$ that is adjacent to $U$, it holds that*

$$Pr[\mathcal{M}(U) \in O] \leq e^\epsilon Pr[\mathcal{M}(U') \in O] + \delta,$$
$$Pr[\mathcal{M}(U') \in O] \leq e^\epsilon Pr[\mathcal{M}(U) \in O] + \delta.$$

We should emphasize that, individual DP does not satisfy the rigorous DP definition, as iDP only focuses on the "point-wise" guarantee of privacy. It protects the neighborhood of a specific dataset of interest, instead of fulfilling DP requirements for all possible adjacent databases. While iDP does not provide the same level of privacy protection as the "worst-case" standard DP, it might be sufficient in certain application scenarios, e.g., data publishing/release, when the procedure is non-interactive and the released dataset is indeed the target that one is interested in privatizing. We discuss it in our work as iDP may provide another direction/option for balancing the trade-off between privacy and utility in practice, based on specific applications.

The intuition of iDP is that, while the standard DP (Definition 2.1) requires indistinguishability between any pair of neighboring databases, in some practical scenarios, the data custodian only holds one "ground truth" database $U$ that needs to be protected. Limiting the scope of the neighborhood could be reasonable in certain practical scenarios. The "indistinguihability" requirement is only cast on $U$ and its neighbors specifically, instead of on any possible dataset. iDP has achieved excellent utility for computing robust statistics at small $\epsilon$ [12].

For the DP algorithms that have been discussed previously in this paper, we first note that, for DP-RP and DP-OPORP, the local sensitivity at any $u \in \mathcal{U}$ equals the global sensitivity. In other words, iDP does not help improve DP-RP and DP-OPORP. Also, we will soon discuss the reason why SignRP can be much better than SignOPORP under iDP. Therefore, we will mainly investigate the SignRP algorithms under iDP. Because the "indistinguihability" requirement of DP is only for $U$ and its neighbors locally, operationally, for SignRP, iDP essentially follows the local flipping probability (Section 4.2 and Figure 1) when computing the perturbation level, which can be much smaller than that required by the standard DP.

We propose two iDP-SignRP methods, based on noise addition and sign flipping, respectively. Both approaches share the same key idea of iDP, that is, many signs of the projected values do not need perturbations. This can be seen from Figure 1, where the "local flipping probability" is non-zero only in the regime when the projected data is near 0 (i.e., $L = 1$ in Algorithm 4). Since in other cases the local flip probability is zero, perturbation is not needed. As a result, out of $k$ projections, only a fraction of the projected values needs to be perturbed. This significantly reduces the noise injected to SignRP and boosts the utility by a very large margin.

## A.2 iDP-SignRP-G by Gaussian Noise Addition

In Algorithm 6, we present the iDP-SignRP-G method for one data vector $u$. We use the "local flipping probability" (e.g., in Figure 1) to choose which projections are perturbed before taking signs. After applying random projection to get $k$ projected values, we do the following steps:

1. We compute noise-indicators $(I_1, ..., I_k)$ for each projected value in $x = \frac{1}{\sqrt{k}} W^T u$ using Algorithm 7. Denote $\mathcal{A} = \{I_j : I_j = 1, j = 1, ..., k\}$ and $N_+ = |\mathcal{A}|$. This is the maximal number of different signs of $x$ and $x' = W^T u', \forall u' \in Nb(u)$.

---

**Algorithm 6:** iDP-SignRP-G (DP-SignRP with Gaussian noise)

**1 Input:** Data $u \in [-1,1]^p$; Privacy parameters $\epsilon > 0$, $\delta \in (0,1)$; Number of projections $k$

**2 Output:** Differentially private sign random projections

**3** Apply RP by $x = \frac{1}{\sqrt{k}} W^T u$, where $W \in \mathbb{R}^{p \times k}$ is a random Rademacher matrix

**4** For every projected value in $x$, compute $(I_1, ..., I_k)$ by Algorithm 7

**5** Let $\mathcal{A} = \{I_j : I_j = 1, j = 1, ..., k\}$ and $\tilde{N}_+ = |\mathcal{A}|$

**6** Compute sensitivity $\triangle_2 = \beta \sqrt{\frac{\tilde{N}_+}{k}}$

**7** Compute $\sigma$ by Theorem 3.2 with $\triangle_2$ and privacy budget $\epsilon$ and $\delta$

**8** Compute $\tilde{s}_j = \begin{cases} sign(x_j), & j \notin \mathcal{A} \\ sign(x_j + G), & j \in \mathcal{A} \end{cases}$, where $G \sim N(0, \sigma^2)$ is iid Gaussian noise

**9** Return $\tilde{s} = [\tilde{s}_1, ..., \tilde{s}_k]$

---

---

**Algorithm 7:** Compute noise-indicator of iDP-SignRP-G for one projection

**1 Input:** Data $u \in [-1,1]^p$; one projected value $z$; adjacency parameter $\beta$

**2 Output:** Indicator $I$ w.r.t. projection $w$ for data vector $u$

**3** $I = 0$

**4 If** $\beta/\sqrt{k} \geq |z|$

**5** $\quad I = 1$

**6 End If**

---

2. We compute the sensitivity $\triangle_2 = \beta \max_{i=1,...,p} \|W_{[i,\mathcal{A}]}\|$, where $W_{[i,\mathcal{A}]}$ denotes the $i$-th row of $W$ indexed at $\mathcal{A}$, which is an $N_+$-dimensional vector.

3. We use the optimal Gaussian mechanism (Theorem 3.2) to compute $\sigma$, with $\triangle_2$ computed above and privacy parameters $(\epsilon, \delta)$.

4. For $j = 1, .., k$, if $j \notin \mathcal{A}$, we take $\tilde{s}_j = sign(x_j)$; if $j \in \mathcal{A}$, we take $\tilde{s}_j = sign(x_j + G)$ where $G \sim N(0, \sigma^2)$ is a Gaussian noise. Finally we output $\tilde{s} = [\tilde{s}_1, ..., \tilde{s}_k]$.

Let's explain the intuition behind DP-SignRP-G. Since a neighboring data vector $u'$ only differs from $u$ in one dimension by at most $\beta$, for each single projection $w$, when $\beta \max_{i=1,...,p} |w_i| \leq |w^T u|$, there is no neighbor $u'$ of $u$ that may change the sign of the projected value of $u$, i.e., $sign(w^T u') \neq sign(w^T u)$. In other words, when $\beta \max_{i=1,...,p} |w_i| \leq |w^T u|$, no noise is needed for this projected value to attain iDP. This is the reason why we call the output of Algorithm 7 a "noise-indicator". Consequently, in step 4 of iDP-SignRP-G it suffices to add Gaussian noise only to those projected values $x_j$ with $j \in \mathcal{A}$, instead of to all $k$ projections as in DP-RP-G-OPT.

**Theorem A.1** (iDP-SignRP-G). *Algorithm 6 is $(\epsilon, \delta)$-iDP for data $u$.*

*Proof.* For a data vector $u$, let $Nb(u)$ be its neighbor set with vector that differs from $u$ by at most $\beta$ in one dimension. Denote $x = \frac{1}{\sqrt{k}} W^T u$ and $x' = \frac{1}{\sqrt{k}} W^T u'$. Let $(I_1, ..., I_k)$ be the noise-indicators from Algorithm 7 and $\mathcal{A} = \{i : I_j = 1\}$, $\tilde{N}_+ = |\mathcal{A}|$. Consider the two sets separately:

- For $j \in [k] \setminus \mathcal{A}$, by the condition $\beta/\sqrt{k} \leq |z|$, we know that $\forall u' \in Nb(u)$, it holds that $sign(x_i) = sign(x_i')$.

- For $j \in \mathcal{A}$, consider the sub-vector $x_{\mathcal{A}}$. Adding iid Gaussian noise to $x_{\mathcal{A}}$ according to Theorem 3.2 with $\triangle_2 = \beta \sqrt{\frac{\tilde{N}_+}{k}}$ ensures the $(\epsilon, \delta)$-DP of $x_{\mathcal{A}}$. By the post processing property of DP, we know that $sign(x_{\mathcal{A}})$ is also $(\epsilon, \delta)$-DP. Thus, for any $Q \in \{-1, 1\}^{N_+}$, we have $Pr(sign(x_{\mathcal{A}}) = Q) - e^\epsilon Pr(sign(x_{\mathcal{A}}') = Q) \leq \delta, \forall u' \in Nb(u)$.

 Combining two parts, we have for any $Q \in \{-1, 1\}^k$,

$$Pr(sign(x) = Q) - e^\epsilon Pr(sign(x') = Q) = Pr(sign(x_\mathcal{A}) = Q) - e^\epsilon Pr(sign(x'_\mathcal{A}) = Q) \leq \delta,$$

for all $u' \in Nb(u)$. By the symmetry of DP (on the sub-vector $x_\mathcal{A}$), we also know that $Pr(sign(x') = Q) - e^\epsilon Pr(sign(x) = Q) \leq \delta$. This proves the $(\epsilon, \delta)$-iDP by Definition A.1. $\square$

## A.3 iDP-SignRP-RR by Randomized Response

---

**Algorithm 8:** iDP-SignRP-RR

---

1 **Input:** Data $u \in [-1, 1]^p$, privacy parameters $\epsilon > 0$, $0 < \delta < 1$, number of projections $k$

2 **Output:** Differentially private sign random projections

3 Apply RP by $x = \frac{1}{\sqrt{k}} W^T u$, where $W \in \mathbb{R}^{p \times k}$ is a random Rademacher matrix

4 For every column in $W$, compute $(I_1, ..., I_k)$ by Algorithm 7

5 Let $\mathcal{A} = \{I_j : I_j = 1, j = 1, ..., k\}$ and $\tilde{N}_+ = |\mathcal{A}|$

6 Compute $\tilde{s}_j = \begin{cases} sign(x_j), & j \notin \mathcal{A} \\ sign(x_j), & j \in \mathcal{A} \text{ with prob. } \frac{e^{\epsilon'}}{e^{\epsilon'}+1} \\ -sign(x_j), & j \in \mathcal{A} \text{ with prob. } \frac{1}{e^{\epsilon'}+1} \end{cases}$ for $j = 1, ..., k$, with $\epsilon' = \epsilon/\tilde{N}_+$

7 Return $\tilde{s}$ as the DP-SignRP of $u$

---

Similar to Section 4, we also have an iDP-SignRP-RR method with pure $\epsilon$-DP guarantee by randomly flipping the signs after SignRP, as summarized in Algorithm 3. After we apply random projection $x = \frac{1}{\sqrt{k}} W^T u$, we call the same procedure as in iDP-SignRP-G to determine set $\mathcal{A}$ representing the projected values that needs perturbation for iDP. For $j \notin \mathcal{A}$, we use the original $\tilde{s}_j = sign(x_j)$. For $j \in \mathcal{A}$, we keep $sign(x_j)$ with probability $\frac{e^{\epsilon'}}{e^{\epsilon'}+1}$ and flip the sign otherwise, where $e^{\epsilon'} = \epsilon/\tilde{N}_+$ with $\tilde{N}_+ = |\mathcal{A}|$.

**Theorem A.2.** *Algorithm 3 achieves $\epsilon$-iDP for data $u$.*

*Proof.* The high-level proof idea is similar to that of Theorem A.1. For $u \in [-1, 1]^p$ let $u'$ be an $\beta$-neighboring data. Let $s = sign(W^T u) \in \{-1, +1\}^k$, $s' = sign(W^T u') \in \{-1, +1\}^k$, and denote $\tilde{s}$ and $\tilde{s}'$ as the randomized output of $s$ and $s'$ by Algorithm 3, respectively. Consider $\mathcal{A}$ in Algorithm 3. By Algorithm 7, we know that for $j \notin \mathcal{A}$, $Pr(\tilde{s}_j = \tilde{s}'_j) = Pr(s_j = s'_j) = 1$, $\forall u' \in Nb(u)$. For projections in $\mathcal{A}$, denote $S = \{j \in \mathcal{A} : s_j \neq s'_j\}$ and $S^c = \mathcal{A} \setminus S$. For any vector $y \in \{-1, +1\}^k$, we further define $S_0 = \{j \in S : s_j = y_j\}$, $S_1 = \{j \in S : s_j \neq y_j\}$, $S_0^c = \{j \in S^c : s_j = y_j\}$ and $S_1^c = \{j \in S^c : s_j \neq y_j\}$. Since the $k$ projections are independent, by composition we have

$$\log \frac{Pr(\tilde{s} = y)}{Pr(\tilde{s}' = y)} = \log \frac{\prod_{j \notin \mathcal{A}} Pr(\tilde{s}_j = y_j) \prod_{j \in S_0^c} \frac{e^{\epsilon'}}{e^{\epsilon'}+1} \prod_{j \in S_1^c} \frac{1}{e^{\epsilon'}+1} \prod_{j \in S_0} \frac{e^{\epsilon'}}{e^{\epsilon'}+1} \prod_{j \in S_1} \frac{1}{e^{\epsilon'}+1}}{\prod_{j \notin \mathcal{A}} Pr(\tilde{s}'_j = y_j) \prod_{j \in S_0^c} \frac{e^{\epsilon'}}{e^{\epsilon'}+1} \prod_{j \in S_1^c} \frac{1}{e^{\epsilon'}+1} \prod_{j \in S_0} \frac{1}{e^{\epsilon'}+1} \prod_{j \in S_1} \frac{e^{\epsilon'}}{e^{\epsilon'}+1}}$$

$$\leq \log \frac{\prod_{j \in S} \frac{e^{\epsilon'}}{e^{\epsilon'}+1}}{\prod_{j \in S} \frac{1}{e^{\epsilon'}+1}} = |S|\epsilon' \leq \tilde{N}_+ \epsilon' = \epsilon,$$

which proves the $\epsilon$-iDP according to Definition A.1. $\square$

The number of projections that requires noise addition $\tilde{N}_+$ is also tightly related to the $P_+(\|u\|, p)$ (Proposition 4.4 and (5)). Particularly, $\tilde{N}_+$ would be small when the data has relatively large norm compared with the change in neighboring data $\beta$. Therefore, both iDP-SignRP methods would have better utility when the data norm is large.

The reduction from $k$ to $\tilde{N}_+$ in iDP not only waives the need to add noise to many projected values, but also requires smaller Gaussian noise or smaller flipping probability for the values that need to be

659  perturbed. Specifically, note that in Algorithm 6, the optimal Gaussian mechanism is deployed with
660  sensitivity $\triangle_2 = \beta\sqrt{\frac{\tilde{N}_+}{k}}$, instead of $\triangle_2 = \beta$ as in (3) for DP-RP-G-OPT.

661  **iDP-SignOPORP**. Similarly, we can also apply iDP to the SignOPORP method. Basically, we only
662  need to replace $x$ in Line 3 in both Algorithm 6 and Algorithm 8 by the OPORP of $u$. However, we
663  note that this iDP-SignOPORP procedure is considerably worse than iDP-SignRP in performance.
664  This is because, by the binning step in OPORP, the average scale of each projected value becomes
665  much smaller. This implies that in Algorithm 7, the magnitude of $z$ would be much smaller, so a
666  lot more projected values will require perturbation, which leads to a utility loss. This illustrates the
667  superiority of SignRP under iDP: since each RP aggregates the whole data vector, SignRP is more
668  robust to a small change in the data. Hence, less noise is needed.

669  ### A.4   Empirical Results on iDP

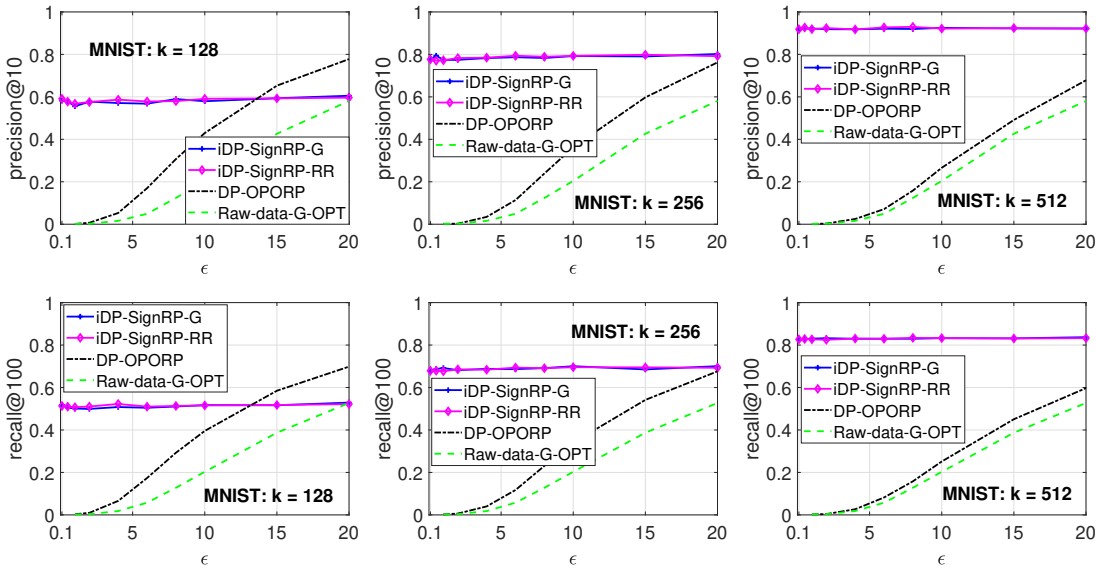

Figure 5: Retrieval on MNIST with iDP-SignRP, $\beta = 1$, $\delta = 10^{-6}$.

670  To demonstrate the empirical gain in utility of iDP-SignRP, we conduct the same set of experiments
671  as in Section 5. Figure 5 reports the precision and recall on MNIST, and Figure 6 presents the
672  SVM test accuracy on WEBSPAM. As we can see, iDP-SignRP achieves very high utility even
673  when $\epsilon < 0.1$. We see that the curves of iDP-SignRP are almost flat. This is because only a small
674  fraction of projected values are perturbed, so the untouched projected values already provides rich
675  information for search and classification. In other words, the experimental results illustrate that the
676  SignRP itself is already very strong in protecting the individual differential privacy. In other words,
677  SignRP itself is already a strong method to protect the privacy of each specific dataset with respect
678  to the individual DP, e.g., in non-interactive data publishing tasks.

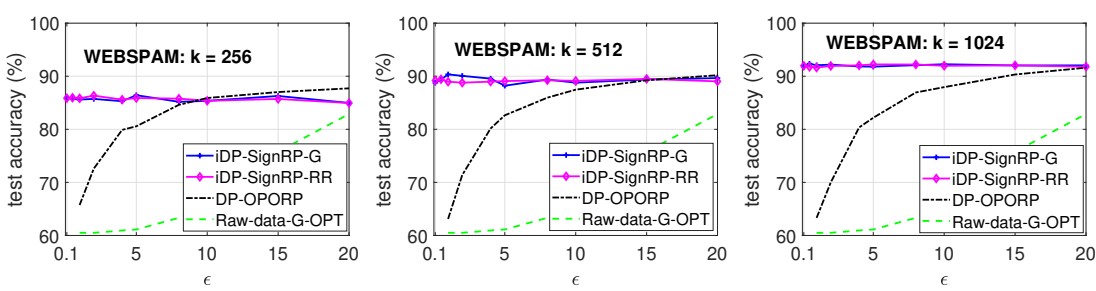

Figure 6: SVM on WEBSPAM with iDP-SignRP, $\beta = 1$, $\delta = 10^{-6}$.

## B Comparison of Different Projection Matrices and the Benefits of Rademacher RP

Besides Gaussian random projection, we can also adopt other types of projection matrices which might even work better for DP. The following distributions of $w_{ij}$ are popular:

- The uniform distribution, $\sqrt{3} \times unif[-1, 1]$. The $\sqrt{3}$ factor is placed here to have $\mathbb{E}(w_{ij}^2) = 1$ by following the convention in the practice of random projections.

- The "very sparse" distribution, as used in [8]:

$$w_{ij} = \sqrt{s} \times \left\{ \begin{array}{lll} -1 & \text{with prob.} & 1/(2s) \\ 0 & \text{with prob.} & 1 - 1/s, \\ +1 & \text{with prob.} & 1/(2s) \end{array} \right. \

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

$$Pr(\max_{i=1,...,p} |X| < |Y|) = \prod_{i=1}^{p} Pr(|X_i| < |Y|).$$

833 By Lemma E.1, among all the possible dependency structures, the above probability reaches its
834 minimum when every $X_i$ is independent of $Y$. Therefore, $Pr(\max_{i=1,...,p} |X| > |Y|) = 1 -$
835 $Pr(\max_{i=1,...,p} |X| < |Y|)$ achieves maximum when $\rho(X_i, Y) = 0$, $\forall i = 1, ..., p$. Since $|X_i|$
836 follows a half-normal distribution with cdf being $\mathrm{erf}(\frac{x}{\sqrt{2}\sigma_x})$, we have

$$Pr(\max_{i=1,...,p} |X_i| \leq t) = \mathrm{erf}\left(\frac{t}{\sqrt{2}\sigma_x}\right)^p = \left[2\Phi\left(\frac{x}{\sigma_x}\right) - 1\right]^p,$$

837 and probability density function $g(x) = 2p[\Phi(\frac{x}{\sigma_x}) - 1]^{p-1} \frac{1}{\sqrt{2\pi}\sigma_x} e^{-\frac{x^2}{2\sigma_x^2}}$. When $Y$ is independent of
838 all $X_i$'s (which gives the upper bound), we have

$$
\begin{aligned}
Pr(\max_{i=1,...,p} |X| > |Y|) &= \int_0^\infty 2p \left[\Phi\left(\frac{x}{\sigma_x}\right) - 1\right]^{p-1} \frac{1}{\sqrt{2\pi}\sigma_x} e^{-\frac{x^2}{2\sigma_x^2}} Pr(|Y| < x) dx \\
&= \int_0^\infty 2p \left[\Phi\left(\frac{x}{\sigma_x}\right) - 1\right]^{p-1} \frac{1}{\sqrt{2\pi}\sigma_x} e^{-\frac{x^2}{2\sigma_x^2}} \mathrm{erf}\left(\frac{x}{\sqrt{2}\sigma_y}\right) dx \\
&= \int_0^\infty 2p[2\Phi(t) - 1]^{p-1}[2\Phi(rt) - 1]\phi(t) dt,
\end{aligned}
$$

839 with a proper change of variables. This gives an upper bound as shown above. $\square$

### E.2 Proof of Proposition 4.4

841 *Proof.* Consider a single Gaussian projection vector $w$ with iid $N(0,1)$ entries. Since $w^T u =$
842 $\sum_{i=1}^{p} u_i w_i$ and each $w_i \sim N(0,1)$, we know that $\begin{pmatrix} \beta w_i \\ x \end{pmatrix} \sim N \begin{pmatrix} \beta^2 & \rho_i \beta \|u\| \\ \rho_i \beta \|u\| & \|u\|^2 \end{pmatrix}$ where $\rho_i =$
843 $\frac{u_i}{\|u\|}$ is the correlation coefficient. Since $|w^T(u - u')| \leq \beta \max_{i=1,...,p} |w_i|$ by Definition 2.2 of
844 $\beta$-neighboring (and more generally, when $\|u - u'\|_1 \leq \beta$), we have

$$Pr(\max_{u' \in Nb(u)} |w^T(u - u')| \geq |w^T u|) = Pr(\beta \max_{i=1,...,p} |w_i| \geq |w^T u|).$$

845 Note that, $\beta \max_{i=1,...,p} |w_i| \geq |w^T u|$ is a necessary condition for the event that there exists a neigh-
846 bor such that $sign(w^T u) \neq sign(w^T u')$. Denote $I = \mathbb{1}\{\beta \max_{i=1,...,p} |w_i| \geq |w^T u|\}$. Applying
847 Lemma 4.2 with $r = \beta/\|u\| \leq 1$ yields

$$\mathbb{E}[I] = Pr(\beta \max_{i=1,...,p} |w_i| \geq |w^T u|) \leq F_{\|u\|,p} = \int_0^\infty 2p[2\Phi(t) - 1]^{p-1}[2\Phi(rt) - 1]\phi(t) dt \quad (21)$$

848 as given by (5). Let $I_j$ be the corresponding indicator function w.r.t. each column in the projection
849 matrix $W$. Denote $N_+ = \sum_{j=1}^{k} I_j$, and by the above reasoning, we know that $|S| \leq N_+$ where $S$ is
850 defined in the theorem. Since the columns of $W$ are independent, $N_+$ follows a $Binomial(k, \mathbb{E}[I])$

distribution with $k$ trials and success probability $\mathbb{E}[I]$ bounded as above. Applying Chernoff's bound on binomial variable (Lemma 4.3), we obtain

$$Pr(N_+ \geq (1+\eta)F_{\|u\|,p}k) \leq \exp(-\frac{\eta^2 F_{\|u\|,p}k}{\eta+2}).$$

Setting the RHS to $\delta$ gives $\eta = \frac{\log(1/\delta)+\sqrt{(\log(1/\delta))^2+8F_{\|u\|,p}k\log(1/\delta)}}{2F_{\|u\|,p}k}$. Therefore, with probability $1-\delta$,

$$N_+(\|u\|,\delta,k,p) \leq F_{\|u\|,p}k + \frac{1}{2}\Big[\log(1/\delta) + \sqrt{(\log(1/\delta))^2 + 8F_{\|u\|,p}k\log(1/\delta)}\Big].$$