# OpenReview forum: "Smooth Flipping Probability for Differential Private Sign Random Projection Methods"
_NeurIPS.cc/2023/Conference — NeurIPS 2023 poster_

### Official Review · Reviewer_hnU4 · 2023-07-05

**Soundness:** 3 good
**Presentation:** 4 excellent
**Contribution:** 3 good
**Rating:** 6
**Confidence:** 3

**Summary:**

This paper develops a series of differentially private (DP) Random Projection and Signed Random Projection algorithms. By using randomized responses and the idea of smooth sensitivity, the new proposed algorithms are shown to outperform previous results while still maintaining the same amount of privacy guarantee.

**Strengths:**

- The idea of using the exact tail probabilities of Gaussian to compute the necessary noise level (Theorem 3.2) is pretty interesting. This might imply that we can improve the empirical performance of most Gaussian Mechanism-based algorithms by using this simple fix.

- "Local perturbation" is also an interesting idea to investigate since a lot of the time, we only need to maintain the privacy for specific sets of datasets rather than for any datasets in the data universe.

- The appendix has plenty of good and thorough discussions that improve the readability and completeness of the paper.

- The proofs and analyses are fairly well-organized and well-written.

**Weaknesses:**

- Seems like we do not have a utility guarantee for the smooth flipping probability algorithms? Since this is the main focus of the paper, the result would be stronger if the author can show that the algorithm performs well both in practice and in theory.

- It would be interesting to see if the new DP-random projection algorithms can perform well in deep learning tasks. For example, if we use the DP-random projection to quantize the gradient, how does that affect our results? I feel like this would better demonstrate the power of the projection algorithms.

- The runtime should probably be reported in the experiments section since it's a fairly important aspect of these types of algorithms.

**Questions:**

- How does the author numerically evaluate $\sigma$ in Theorem 3.2? Do we have a closed form solution for that?

- I'm also curious how the author compute $N_{+}$ in practice since it looks like a pretty complicated bound?

- Since Proposition 4.4 suggests that $N_{+}$ is smaller when the data norm is large compared to $\beta$, can we just scale up the data in practice?

---

> ### Author Rebuttal · Authors · 2023-08-10
>
> Dear Reviewer hnU4:
>
> Thanks for your feedback. We appreciate your interests in our work and the ideas behind the algorithms. We also hope that the theoretical and empirical results in our paper (the adoption of optimal Gaussian mechanism and the smooth flipping mechanism) could be helpful to the community.
>
> W1. Thanks for the question. In principle, we can derive some results for DP-SignRP-RR-smooth like the estimation variance in Theorem 4.5. However, this would be very complicated, since by using the smooth flipping probability, we need to compute the probability that the projected value falls in different "regions". We can use a similar calculation as in Lemma E.1, to express those probabilities as some tedious integrals. The final expression would be even more complicated when considering the interactions between the DP-SignRP-RR-smooth sketches of two data vectors. Therefore, in the submission, we mainly focused on elaborating the algorithm design and intuition, and the empirical evaluations. We are happy to add the analytical formula to the paper if suggested by the reviewer.
>
> W2. In this paper, we focus on the more fundamental aspects of the differential privacy of RP based algorithms, since many details and ideas (e.g., the smooth flipping probability) have not been addressed before. In future works, it would be interesting to apply these algorithms to more tasks/areas, such as deep learning as you kindly suggested. Thank you.
>
> W3. In general, since RP and OPORP only involves a matrix-matrix and vector-matrix multiplication which have very efficient implementations in the computing softwares, generating the sketches (or signed sketches) takes very little proportion of time compared with the time for search and model training. The flipping mechanisms can also be efficiently implemented by properly generating a binary mask matrix according to the flipping probabilities. The OPORP based algorithms are even faster since to generated $k$ projected values, we only need to touch each dimension of the data vector once. Thus, in the experiments we mainly focused on the performances of the proposed methods. If the reviewer is still interested in the overall running times, we are happy to share the more detailed running time evaluations.
>
>
> Q1. We numerically solved the (one-dimensional) nonlinear equation for $\sigma$, for different $\triangle_2, \epsilon$, and $\delta$ values. There is no closed-form solution for this problem. We have provided the detailed numerical values for $\sigma$ as a plot, in a more detailed technical report. We are happy to include such a plot in the supplementary material if Reviewers feel it is necessary. Thank you.
>
> Q2. As stated in Theorem 4.4, we first compute $F_{\|u\|,p}=P_+(\frac{\beta}{\|u\|},p)$ as in Eq. (5), Lemma 4.2. Then we plug $F_{\|u\|,p}$ into Eq. (6). So basically we only need to do an Gaussian integration (5).
>
> Q3. Note that in our paper, we assumed that $u\in [-1,1]^p$ and defined the distance between two neighbors to be $\beta$. Alternatively, as the reviewer mentioned, we can scale the data domain to $[-C, C]$ with some $C>1$. However, in this case, to attain equal-strength privacy protection, the adjacency parameter should also be scaled to $C\beta$. Thus, the ratio $\beta/\|u\|$ would remain the same after scaling. Hence, simply scaling up the data would not help enhance privacy, unless we relax the privacy protection requirement (e.g., do not scale up $\beta$ accordingly). We hope this answers the question. Thank you.
>
> Thanks again for your feedback on our work. We hope our reply answers your questions adequately.

---

> > ### Comment · Reviewer_hnU4 · 2023-08-19
> >
> > Thanks for the response! I think my questions are well-addressed. I will keep my score.

---

### Official Review · Reviewer_TuXU · 2023-07-06

**Soundness:** 4 excellent
**Presentation:** 3 good
**Contribution:** 3 good
**Rating:** 7
**Confidence:** 4

**Summary:**

The authors study the important problem of preserving privacy in matrix data. They study a family of fundamental operations on data which is Random Projection. This is useful for dimensionality reduction, NN search etc. Based on RP one can also obtained signed RP (SRP) which are useful (as in  SimHash) to find near duplicates and estimate similarity of vectors, etc. The authors present DP algorithms for this family of applications (RP, SRP). Their algorithm improve over prior work ([47]) showing methods of reducing the amount of noise using a series of techniques including smooth sensitivity and better upper bounds on sensitivity. Their experiments show improved results.
All in all this is an interesting paper on improving the accuracy of an important building block in data analysis with privacy.

**Strengths:**

+ Important problem : computing with privacy random projection based sketches
+ Improved empirical results for a practical algorithm
+ theoretically sound and non-trivial results

**Weaknesses:**

- It would be interesting to know how much head room is there if the projection matrix itself is kept secrete. There is no comparison with methods that omit the matrix W.


**Questions:**

1) Can you discuss more the comparison with methods like Blocki et al that keep W secrete (or even provide empirical results for comparing which such methods)? Is it possible to improve the results significantly?

---

> ### Author Rebuttal · Authors · 2023-08-10
>
> Dear Reviewer TuXU:
>
> We appreciate your encouraging feedback on our theoretical and empirical results, and the significance of the DP sketching problem. We too believe that DP sketching methods (e.g., the RP-based algorithms in our paper) are interesting, important, and could have broad applications.
>
> The DP-RP method under the setting where the projection matrix $W$ is also private/random was studied in [1]
>
> [1] Blocki et al., The johnson-lindenstrauss transform itself preserves differential privacy, FOCS 2012.
>
> [1] argued that when the singular values of the data matrix are all large enough, the randomness of RP itself provides DP. While the authors studied a slightly different problem of publishing covariance matrix through RPs, their analysis also implies that using the same Gaussian noise addition mechanism as in DP-RP, the Gaussian noise requires $\sigma=O(\sqrt{k}/\epsilon)$ when $W$ is private. Asymptotically, this is the same as the noise level with known $W$ when $\epsilon\rightarrow 0$, and is smaller than $O(\sqrt k/\sqrt\epsilon)$ with known $W$ when $\epsilon\rightarrow \infty$ (see our Remark 3.1). Theoretically, we find that the analysis in [1] is quite tight, though no lower was proved.
>
> Moreover, for SignRP, we are not aware of any result/analysis under the private $W$ setting. The analysis seems challenging, but we believe this could be an interesting problem to explore.
>
> Again, thanks for your review of our work.

---

> > ### Comment · Reviewer_TuXU · 2023-08-14
> >
> > Thanks to the author for the response. My question is well-addressed. I will keep my score.

---

### Official Review · Reviewer_bvWd · 2023-07-06

**Soundness:** 4 excellent
**Presentation:** 4 excellent
**Contribution:** 4 excellent
**Rating:** 7
**Confidence:** 4

**Summary:**

The paper introduces a series of differential privacy (DP) random projections (RP)  algorithms based on the robustness of the sign flipping probability in RP. This approach is promising as it seeks to enhance the existing methods in both theoretical guarantees and empirical performance.

**Strengths:**

1. The analysis of the probability of smooth sign flipping is a novel contribution, offering a fresh perspective for DP RP algorithms.

2. The presentation of the paper is generally clear, providing a concise explanation of the methodology and algorithms developed. The authors effectively explain the motivation behind their research and the rationale for leveraging sign flipping probability.

3. The experimental evaluation of the proposed algorithms is adequate, highlighting the improvements achieved by DP-SignOPORP and iDP-SignRP over existing algorithms in the standard differential privacy setting. The comparison with other methods in the literature provides valuable insights into the performance of the proposed algorithms.

**Weaknesses:**

NA

**Questions:**

Could you provide more intuition on the scale of the smoothness in terms of sign flipping probability?

**Limitations:**

The authors adequately addressed the limitations

---

> ### Author Rebuttal · Authors · 2023-08-10
>
> Dear Reviewer bvWd:
>
> Thanks for your feedback on our paper.
>
> As we mentioned at line 195, The concrete exponentially decaying form of the flipping probabilities in Algorithm 4 is leads to the "smoothness" of the flipping probability: for any neighboring $u$ and $u'$, $P^{(u)}\leq e^{\epsilon/k}P^{(u')}$. By re-arranging we see that $P^{(u)}/P^{(u')}\leq e^{\epsilon/k}$. In the proof, this bound the key to guarantee that each projected value takes $\frac{\epsilon}{k}$ privacy budget, leading to the $\epsilon$-DP. Figure 4 plots the smooth flipping probability, the standard flipping probability (global upper bound), and the local flipping probability (which is non-DP). With the exponential decay, smooth flipping probability is much smaller than the worst-case analysis.
>
> We are glad that you are interested in the smooth flipping probability, which is a novel idea for privatizing the SignRP and SignOPORP. We expect that this idea could be helpful for more DP algorithm designs. We hope now the intuition and design of the smooth flipping mechanism is more clear. Thanks again for your review.

---

> > ### Comment · Reviewer_bvWd · 2023-08-14
> > **Thank you for the response.**
> >
> > Thanks to the author for the response. My question is well-addressed. I will keep my score.

---

### Official Review · Reviewer_ywEG · 2023-07-07

**Soundness:** 2 fair
**Presentation:** 1 poor
**Contribution:** 2 fair
**Rating:** 4
**Confidence:** 3

**Summary:**

This paper studies the problem of computing differentially private sketches of high dimensional vectors via random projections. The contributions are:
1. Algorithm DP-RP-G-OPT: An algorithm that uses the Gaussian mechanism to privatize random projections using Gaussian and Rademacher matrices, but adds more carefully calibrated noise by using the CDF of the Gaussian distribution as opposed to the usual tail bounds. It outputs a k-dimensional DP sketch of a p dimensional input vector, it seems to be working in the local DP setting as it operates on a single data vector.
2. Algorithm DP-OPORP: An algorithm that appends the OPORP mechanism of Li and Li ([53] in the paper) with the Gaussian mechanism to produce a DP variant of the former. It produces a k dimensional sketch of the p dimensional input vector but uses only $p$-many multiplication operations instead of $kp$-many.
3. Algorithm DP-SignRP: An algorithm that produces a 1-bit sketch of the input vector in a differentially private way. The algorithm only accepts input vectors with some lower bound $m$ on the norm which must be passed as an argument. They also use this algorithm to produce a DP sign estimator.
4. Algorithms DP-SignRP-RR and DP-SignRP-RR-Smooth: The authors aim to produce an improved version of DP-SignRP by leveraging the stability of the sign of the projected output under small changes in the input. They also use the binning of OPORP to reduce the sensitivity of the algorithm to the input so as to be able to add less privatizing noise.

The authors go on to test their methods empirically on similarity search and classification problems.

**Strengths:**

1. The authors want to leverage the stability of the sign operator under random projects to boost the privacy guarantee over what one might get naively, in some sense they want to appeal to the local sensitivity of the mapping when the input vector is far away from the decision boundary. This idea may be fruitful.
2. The authors conduct some experiments to test their methods.

**Weaknesses:**

1. The privacy model here is not clear - the authors define and talk about differential privacy in the central model (where the stability guarantee holds over adjacent data sets) but their algorithms take as input only one data point. It seems that instead all the guarantees are meant to hold in the non-interactive local model of differential privacy.
2. The methods DP-RP-G-OPT and DP-OPORP are not very novel - the first achieves no asymptotic improvements over prior work and seems to improve the performance by some unspecified constant. The second simply appends the OPORP algorithm from prior work by the Gaussian mechanism.
3. The significance DP-SignRP-RR is not clear - there is a lower bound $m$ on the input vector.
4. The quality of the writing needs to be improved. The most novel contribution seems to be DP-Sign-OPORP-smooth-RR which is relegated to the end.

**Questions:**

1. Can you please elaborate on the model of differential privacy that you are operating in? What is the significance of this for the CountSketch application that you describe in the beginning?
2. Why do we need to assume a lower bound on the norm of the input vector for DP-SignRP-RR? Does this impact its applicability/parameter regimes? How should we set the value of $m$? In particular it seems like $m<\sqrt{p}$ is necessary to avoid the domain becoming empty.

**Limitations:**

This is a theoretical work and I do not feel that potential societal impact needs to be discussed here.

---

> ### Author Rebuttal · Authors · 2023-08-10
>
> Dear Reviewer ywEG:
>
> Thanks for your feedback on our submission.
>
> W1 and Q1.  As Reviewer mentioned, the proposed private sketching algorithms are designed to protect attribute-level privacy. That is, our method prevents the identification of a change in any element in the vector in the sense of DP. This is becauseRP is a "sample-wise" algorithm---RPs are concrete representations of the original data and are directly used for downstream tasks for search and learning. In fact, due to this nature of sketching methods, this attribute/item-level privacy setup is considered by many prior papers on DP sketching, for example,
>
> Kenthapadi et al., Privacy via the johnson-lindenstrauss transform. J. Priv. Confidentiality 2013.
>
> Nina Mesing Stausholm, Improved differentially private euclidean distance approximation. PODS 2021.
>
> Zhao et al., Differentially private linear sketches: Efficient implementations and applications. NeurIPS 2022.
>
> Smith et al., The flajolet-martin sketch itself preserves differential privacy: Private counting with minimal space. NeurIPS 2020.
>
> Dickens et al., Order-invariant cardinality estimators are differentially private. NeurIPS 2022.
>
> In a centralized setting with a database consisting of $n$ data points (vectors), we apply our algorithm to all $n$ data vectors. The privacy statement is: *"an adversary cannot detect a small change in any user's data vector from the user's DP-RP or DP-SignRP samples."* On the other hand, as Reviewer kindly pointed out, our algorithm can also be applied to "local" DP settings where each local user owns one data vector.
>
> To answer Q1 about the application scenarios, our algorithm and privacy protection (or, more generally, DP sketching methods) could be very useful when, for instance, an organization needs to share sensitive data like profiles or gene sequences data for machine learning tasks. Basically, the DP sketches can be used in all the applications where RP (or count sketch) are used. We have introduced and cited many of them in the paper.
>
> If the reviewer hopes to see additional concrete applications, one example is that the bioinformatics community releases sets of 1000 sketches for all known genomes on a regular basis [1]. The sketches are used in various ML tasks like clustering and distance estimation, etc. In this application, our proposed algorithms based on DP-RP, DP-OPORP and their signed versions can all be applied, which protects the privacy of gene expressions of each patient.
>
> [1] Mash: fast genome and metagenome distance estimation using MinHash, Ondov et al., Genome Biology, 2016
>
> W2. DP-RP and DP-OPORP algorithms: In our paper, we hope to provide a complete picture of DP sketching using RP-based algorithms and a step-by-step derivation of the signed RP and OPORP sketches (i.e., DP-SignRP, DP-SignOPORP)
> from DP-RP. We include these algorithms in this submission since they are all highly related. We have tried our best to optimize the space and organization.
>
> The focus (and the title) of our paper is "Smooth Flipping Probability for Differential Private Sign Random Projection Methods".  DP-RP and DP-OPORP are important baselines in our paper to compare with signed methods. This is why we spent about 1 page to review prior works on DP-RP, and proposed an improvement. Section 3 play two roles: (1) adopting the optimal Gaussian mechanism to RP and OPORP has not been proposed and evaluated in the literature before, so we hope this integration and the empirical evaluations could be useful to the community; (2) we aim to improve the baselines of (full-precision) DP-RP and DP-OPORP, to make a fair comparison with the proposed DP-SignRP and DP-SignOPORP methods.
>
> W3 and Q2. In DP-SignRP-RR, which is built upon the classic randomized response (RR) strategy, $\|u\| \geq m$ is assumed for better utility. In fact, if we allow $m=0$, then DP-SignRP-RR reduces to the most naive RR approach with flipping probability $1/(e^{\epsilon/k}+1)$, where $k$ is the number of projected samples. If we know that the data norm is lower bounded by $m>0$, then we can reduce the flipping probability, as in our analysis. In practice, this is quite common. One example is the Genome data (e.g., gene expression microarrays) for which sketching and hashing are widely used. The gene expression data are usually dense with large norms. In this case, we can have some large $m$ numbers.
>
> *"How should we set the value of $m$?"*
>
> Answer: $m$ is not "set" but depends on the data (we assumed all the data points have norm $\geq m$). In our experiments, we let $m$ be the smallest norm among all data points in the dataset. We can also determine it if we have some prior knowledge of the data.
>
> $m$ is just a lower bound on the data point's $l_2$ norm, so there is no constraint for it. It can range from 0 (for all empty vector) to $p$ (when all data entries equal to $\pm 1$).
>
> Finally, we highlight that in our proposed DP-SignRP-RR-smooth with smooth flipping probability, the need of $m$ is waived.
>
> W4. Thanks for the suggestion. We organized Section 4 in this way to show the whole path towards our proposed algorithms, and why and how they improve the classic approach (DP-SignRP-RR). Particularly, the differential privacy of SignRP and SignOPORP has not been studied in the literature before, so we hope to conduct a thorough investigation. Thus, in Section 4.1, we start with the standard strategy to privatize SignRP using the classic randomized response (RR) technique, which requires some non-trivial probability calculations and privacy analysis for the algorithm design. Then in Section 4.2, we propose the novel idea of smooth flipping probability to improve the standard DP-SignRP-RR method, and further proposed DP-SignOPORP-smooth. While the smooth flipping probability is what we recommend eventually, both algorithms present novel analysis and ideas to the literature.
>
> Thanks again for your review. We hope our response has well answered your questions.

---

> ### Author Response · Authors · 2023-08-15
> **Any further questions? Thanks.**
>
> Dear Reviewer ywEG:
>
> Again, we appreciate your time and effort in reviewing our work. In our rebuttal, we have clarified the organization of our paper, background of DP hashing/sketching, the privacy setup, and some more concrete examples of the application scenario. We also hope that now it is more clear that the lower bound $m$ can help reduce the flipping probability of the classic DP-SignRP-RR method, and is not needed by our proposed main algorithm with smooth flipping probabilities.
>
> Since the author-reviewer discussion period is approaching the end, we are wondering if there are more questions that we can help address? Please kindly let us know if further clarification is needed. Thank you.

---

> > ### Comment · Reviewer_ywEG · 2023-08-16
> >
> > Thank you for your rebuttal. It seems that if in the input to algorithm 3 the choice of $m$ is data-dependent, then it must also be privatized prior to usage (ideally by the Gaussian mechanism as you are working in the approximate DP setting). Also I appreciate the clarification about the privacy setting. I think overall my issues with the writing remain, but I will change my score from a 3 to a 4.

---

> > > ### Author Response · Authors · 2023-08-19
> > >
> > > Dear Reviewer ywEG,
> > >
> > > Thanks for your reply. Indeed, our paper did not assume $m$ is data-dependent. In other words, in DP-SignRP-RR, we assumed that $m$, the lower bound on data norm, is a public information.
> > >
> > > Also, thank you for letting us know that you still have concern's on the writing, which, based on your review,  is that our contribution on "smooth flipping probability" did not appear early enough in the paper. As we have explained,  we must introduce in details the previous works (otherwise we would have also received complaints about writing that we did not well explain the previous works which are tightly related to our contribution).
> > >
> > > Our title is "Smooth Flipping Probability for Differential Private Sign Random Projection Methods".  We expected that, this would allow readers to directly see that one of our main contributions is about "smooth flipping probability".
> > >
> > > On page 2, in the contribution section,  we elaborated our contributions on "smooth flipping probability" starting in Line 61, with quite some details. Also, we counted, the main paper in our submission had the word "smooth" 44 times.
> > >
> > > Of course, your criticism is well-taken. We will mention "smooth flipping probability" earlier in the paper.
> > >
> > > On the other hand, this (i.e., not mentioning "smooth flipping probability" early enough) appears to be a minor issue and could be fairly easily fixed. We sincerely hope that Reviewer could take this into account when assigning the final grade. Thanks so much.

---

### Decision · Program_Chairs · 2023-09-21

**Decision:**

Accept (poster)

**Comment:**

Reviewers bvWd, TuXU, and hnU4 are all in favor of accepting the paper, arguing that the analysis is interesting and novel, some of the proposed methods have promising empirical results, and that the proofs are well organized and written. Reviewer ywEG raises two weaknesses: 1. the privacy model is not clear, and 2. the contributions of the paper are unclear.

I agree that the privacy model was unclear in the original submission, but I feel that the authors adequately clarified during the discussion period. Section 2 should be updated to clarify that the paper studies **attribute-level** privacy, where adjacent datasets differ in the value of one attribute in one row. In particular, on line 83 the authors write "In this work, we follow the convention in the literature that neighboring databases $U$ and $U'$ only differ in one row." While this is technically true, this sentence is confusing because it sounds like neighboring datasets may differ on an entire row, which is not the case in the paper. I would suggest highlighting the fact that this is different from the majority of the DP literature, but common for papers on DP sketching.

The second concern of ywEG is that several of the mechanisms from Section 3 are not new, and it is unclear what the main contributions of the paper are. It may be worthwhile including a brief discussion in Section 3 to help readers identify the main contributions. During the reviewer discussion I asked the other reviewers if they had concerns about the clarity of the paper given reviewer ywEG's review, and reviewers bvWd and TxXU both said that they had no issues.

Given the above minor additions, I believe that the paper is above the bar for NeurIPS.